# Biochemical and structural insights of multifunctional flavin-dependent monooxygenase FlsO1-catalyzed unexpected xanthone formation

Chunfang Yang[1,2,3,4,5], Liping Zhang[1,2,3,5], Wenjun Zhang[1,2,3,4], Chunshuai Huang[1,4], Yiguang Zhu[1,2,3,4], Xiaodong Jiang[1,4], Wei Liu[1], Mengran Zhao[1,4], Bidhan Chandra De[1,4] & Changsheng Zhang [1,2,3,4] ✉

Xanthone-containing natural products display diverse pharmacological properties. The biosynthetic mechanisms of the xanthone formation have not been well documented. Here we show that the flavoprotein monooxygenase FlsO1 in the biosynthesis of fluostatins not only functionally compensates for the monooxygenase FlsO2 in converting prejadomycin to dehydrorabelomycin, but also unexpectedly converts prejadomycin to xanthone-containing products by catalyzing three successive oxidations including hydroxylation, epoxidation and Baeyer-Villiger oxidation. We also provide biochemical evidence to support the physiological role of FlsO1 as the benzo[b]-fluorene C5-hydrolase by using nenestatin C as a substrate mimic. Finally, we resolve the crystal structure of FlsO1 in complex with the cofactor flavin adenine dinucleotide close to the "in" conformation to enable the construction of reactive substrate-docking models to understand the basis of a single enzyme-catalyzed multiple oxidations. This study highlights a mechanistic perspective for the enzymatic xanthone formation in actinomycetes and sets an example for the versatile functions of flavoproteins.

The xanthone scaffold is known as a privileged structure in drug discovery[1,2], for the γ-pyrone containing tricyclic scaffold can devote themselves to diverse pharmacological properties[3], such as anti-inflammatory, anti-oxidant, anti-microbial, and anti-tumor activities[4,5]. Since the report of the xanthone compound albofungin four decades ago[6], numerous xanthones have been isolated from higher plants, lichens, and microorganisms[2,7], e.g., the fungi-derived demethylsterigmatocystin (DMST, **1**)[8,9], neosartorin (**2**)[10], agnestin A (**3**)[11] and nidulaxanthone (**4**)[12], and the actinomycete-originated xantholipin (**5**)[13] and

monacyclione G (**6**) (Fig. 1)[14]. Diverse methods have been developed for the chemical synthesis of xanthones (Supplementary Fig. 1)[15]. However, most biosynthetic studies of xanthone-containing compounds have been limited to genetic analysis and isotope incorporation studies[8–11]. More recently, several excellent studies have demonstrated the enzymatic formation of the xanthone scaffold, such as, the bifunctional cytochrome P450 enzyme HpCYP81AA1-catalyzed formation of 1,3,7-trihydroxyxanthone in the plant *Hypericum calycinum*[16], multiple enzymes-mediated oxidative transformation of

[1]Key Laboratory of Tropical Marine Bioresources and Ecology, Guangdong Key Laboratory of Marine Materia Medica, China-Sri Lanka Joint Center for Education and Research, South China Sea Institute of Oceanology, Chinese Academy of Sciences, Guangzhou 510301, China. [2]Southern Marine Science and Engineering Guangdong Laboratory (Guangzhou), 1119 Haibin Road, Nansha District, Guangzhou 511458, China. [3]Sanya Institute of Ocean Eco-Environmental Engineering, Yazhou Scientific Bay, Sanya 572000, China. [4]University of Chinese Academy of Sciences, 19 Yuquan Road, Beijing 100049, China. [5]These authors contributed equally: Chunfang Yang, Liping Zhang. ✉e-mail: czhang2006@gmail.com

**Fig. 1 | Typical xanthone-containing natural products and the proposed oxidative steps in the biosynthesis of atypical angucyclines (yellow background).** The xanthone scaffold was highlighted in red. The boxed structures showed typical products characterized from the FlsO1-catalyzed reaction with PJM (**8**).

chrysophanol into the xanthone blennolide A in fungi[17], and a single flavoprotein monooxygenase (FPMO[18]) XanO4-catalyzed biosynthesis of the xanthone core in xantholipin (**5**) in actinomycetes[13]. However, the exact mechanism for the enzymatic construction of the xanthone scaffold remains elusive in fungi and actinomycetes.

Fluostatins (FSTs), such as FST C (**7**, Fig. 1), belong to benzofluorene-containing atypical angucyclines with enzyme inhibition and cytotoxic activities[19,20]. The structure diversity of FST-related angucyclines and angucyclinones was largely expanded in recent years by natural isolation[21–26], heterologous expression of the biosynthetic gene cluster (BGC)[27–30], and manipulation of the biosynthetic genes[31–33]. Multiple flavoenzymes were identified in the FST BGC from the marine-derived *Micromonospora rosaria* SCSIO N160 (*fls*)[28]. The monooxygenase FlsO2 was biochemically characterized to efficiently convert prejadomycin (PJM, **8**) to dehydrorabelomycin (DHR, **9**) through CR1 (**10**)[28]. Interestingly, it remained enigmatic why the production of DHR (**9**) was still observed in the *flsO2*-inactivation mutant of *M. rosaria* SCSIO N160[28]. Biosynthetic studies on the FST-related atypical angucycline kinamycin A (**11**) (Fig. 1) have demonstrated that the monooxygenase AlpJ (a homolog of FlsG) catalyzed an oxidative B-ring cleavage and contraction to convert DHR (**9**) to the benzo[*b*]-fluorene intermediate (**12**), which was further hydroxylated to hydroquinone-kinobscurinone (**13**) by the flavoenzyme AlpK (Fig. 1)[34–36]. We also

performed the in vivo characterization of the FPMO FlsO1 as a C-5 hydroxylase that putatively converted **12** to **13** (Fig. 1)[32], functionally equivalent to its homologous enzyme AlpK. However, the in vitro biochemical characterization of AlpK and FlsO1 was hampered by the inavailability of the highly instable substrate **12**, which was prone to become a dimer (or a trimer), or to conjugate with other reactive species via quinone methide-mediated C–C coupling reactions[32,34,35]. Interestingly, a potential substrate mimic nenestatin C (NEN C, **14**), bearing the same carbon scaffold as **12**, was recently isolated as a biosynthetic intermediate of nenestatin A (**15**) (Fig. 1), an atypical angucycline discovered from marine-derived *Micromonospora echinospora* SCSIO 04089[37–39].

In this work, we show that FlsO1 catalyzes an unexpected conversion of PJM (**8**) to multiple products including DHR (**9**), two xanthone-containing products **16** and **17**, and a C-ring opened compound **18** (Fig. 1). The mechanism of FlsO1-catalyzed xanthone formation is demonstrated by characterizing the reaction intermediates/shunt products, to support the involvement of three successive oxidations including a hydroxylation, an epoxidation and a Baeyer-Villiger oxidation. The physiological function of FlsO1 as the benzo[*b*]-fluorene C5 hydroxylase is indirectly confirmed by using NEN C (**14**) to mimic the natural substrate. Finally, the crystal structure of FlsO1 in complex with the cofactor flavin adenine dinucleotide (FAD) is resolved to gain

insights into the structural basis of a single enzyme-catalyzed multiple oxidations.

## Results and discussion

### Discovery of FlsO1-mediated unexpected formation of xanthones

We have previously proposed that homologous oxygenases encoded in the *fls* BGC might complement the function of FlsO2 to convert PJM (**8**) to DHR (**9**) due to the remaining production of DHR (**9**) in the Δ*flsO2* mutant (Supplementary Table 1)[28]. To test this idea, another 4 oxygenases FlsO1, FlsO3, FlsO4, and FlsO5, exhibiting 56.8, 51.3, 58.3, and 56.9% amino acid sequence identity to FlsO2[28], respectively, were overproduced in *Escherichia coli* BL21(DE3) and purified to near homogeneity (Supplementary Fig. 2 and Table 2). Subsequently, FlsO1, FlsO3, FlsO4, and FlsO5 were independently incubated with PJM (**8**) and NADPH, using FlsO2 as the positive control. As expected, FlsO2 catalyzed the conversion of PJM (**8**) to DHR (**9**) through CR1 (**10**)

(Fig. 2a, traces i–iii). FlsO3, FlsO4, and FlsO5 showed no activities with PJM (**8**) (Fig. 2a, traces iv–vi). Interestingly, FlsO1 was found to transform PJM (**8**) to DHR (**9**), together with three additional products **16**–**18** that were distinct from the intermediate CR1 (**10**) in the FlsO2 reaction (Fig. 2a, traces vii–x). Notably, DHR (**9**) was produced in buffers of pH values lower than pH 7, with **18** as the major product (Fig. 2a). Whereas, **16** was observed to be the dominant product in buffers of pH values higher than pH 7 (Fig. 2a). In contrast, in assays using the FlsO1 homologous enzymes AlpK and Nes26 from the kinamycin and nenestatin biosynthetic pathways[34,37], respectively, no conversions of **8** were observed (Supplementary Fig. 3).

A scaled up reaction was then performed with FlsO1 and PJM (**8**) to allow the isolation of the three additional products **16**–**18** (Fig. 1) for structure elucidation by extensive spectroscopic analysis of the HRE-SIMS and NMR data (Supplementary Figs. 4–24 and Tables 3 and 4). Unexpectedly, both **16** and **17** were determined to be xanthone-containing compounds, designated fluoxanthones A (**16**) and B (**17**),

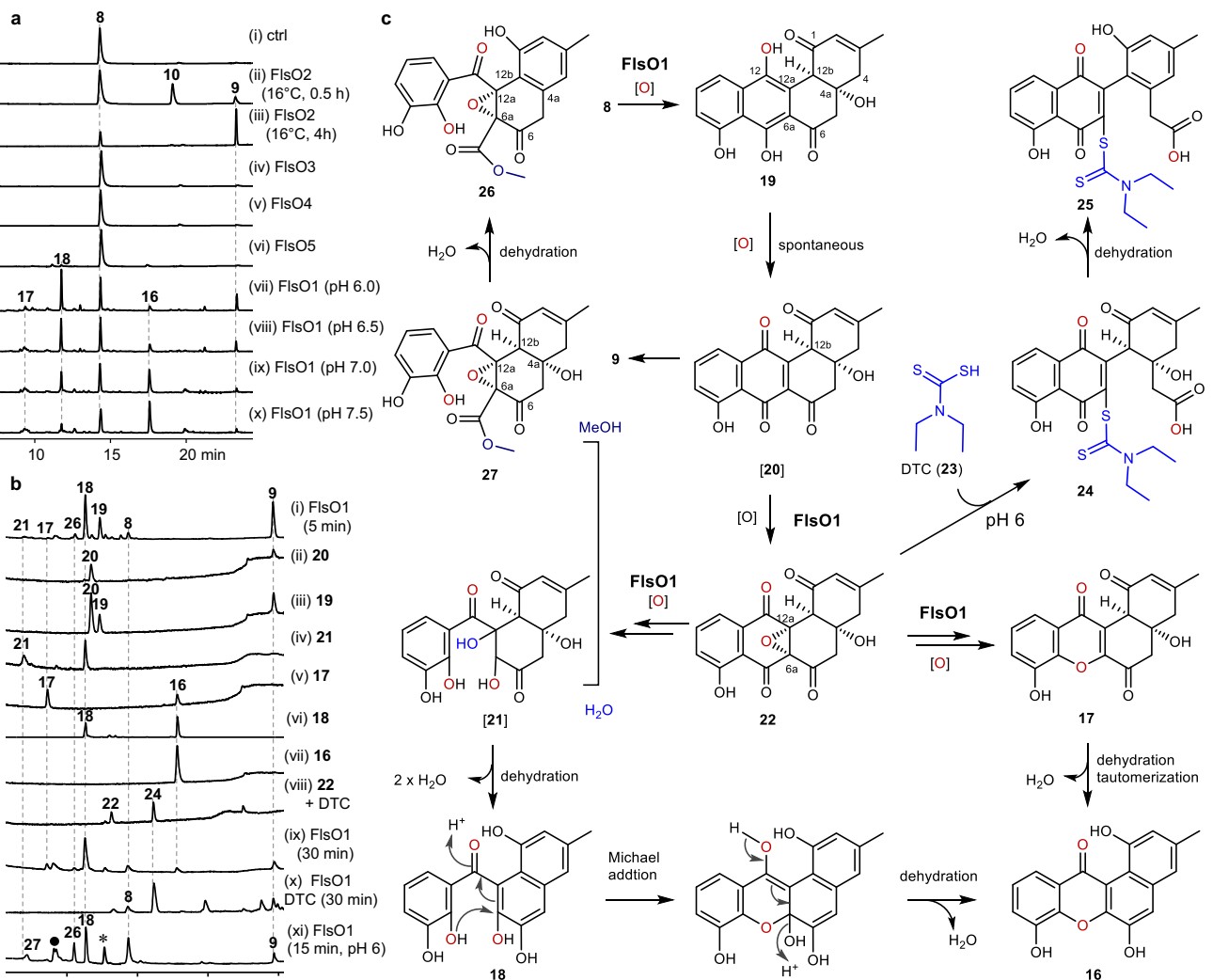

**Fig. 2 | Biochemical characterization of FlsO1-catalyzed formation of 16 from PJM (8). a** HPLC analysis of enzyme assays. The assays were performed by incubation of 200 μM **8** in the presence of 2 mM NADPH: (i) control (no enzyme); (ii, iii) 5 μM FlsO2; (iv) 10 μM FlsO3; (v) 10 μM FlsO4; (vi) 10 μM FlsO5; and (vii–x) 10 μM FlsO1. The reactions were performed in PBS buffers (50 mM) at 16 °C for 0.5 h (ii) and 4 h (iii) for the detection of **10**, or at 30 °C for 30 min (iv–x). The pH values for PBS buffers were pH 7.0 (i–vi and ix), pH 6.0 (vii), pH 6.5 (viii) and pH 7.5 (x), respectively. **b** HPLC analysis of short-time FlsO1 assays and the putative reaction intermediates. (i) FlsO1 reactions with PJM (**8**) for 5 min in PBS buffers (50 mM, pH

7.0) at 30 °C, and the immediate analysis of the spontaneous conversion of putative reaction intermediates after collection from analytical HPLC analysis in trace i; (ii) **20**; (iii) **19**; (iv) **21**; (v) **17**; (vi) **18**; (vii) **16**; (viii) the reaction of **22** with DTC (**23**); the reactions for **8** with FlsO1 in the absence (ix) or the presence (x) of DTC (**23**) for 30 min; (xi) the FlsO1 reaction with **8** in 50 mM phosphate buffer (pH 6.0) for 15 min. Unidentified compounds were indicated with a full black circle and a star. **c** A scheme for the FlsO1 reaction proposed on the characterization of intermediates. Uncharacterized products were bracketed.

respectively. The structure of **18** was elucidated as a derivative of C-ring opened DHR, designated fluoxanol. The absolute configuration of **17** was assigned as 4a*R*,12b*R* by comparison of the experimental ECD spectra of **17** and **8** (Supplementary Fig. 25).

### Characterization of putative intermediates of the FlsO1 reaction with PJM (8)

It was highly unexpected to find **16** and **18** as the major products of the FlsO1 reaction with **8**. For better understanding their formation, a time course assay of FlsO1 (10 µM) with **8** was performed to search for putative intermediates. LC-MS analysis of a reaction solution sampled at 5 min showed the presence of mass ions for products DHR (**9**), and **16**–**18** (Fig. 2b, traces i; Supplementary Fig. 26). In addition, the molecular weight (MW) were determined for several putative intermediates **19** ($m/z$ 339.7 [M − H]$^-$), **20** ($m/z$ 337.8 [M − H]$^-$), **21** ($m/z$ 363.7 [M + H]$^+$), and **22** ($m/z$ 355.7 [M + H]$^+$) (Supplementary Fig. 27), respectively. These intermediates were individually collected from an analytical HPLC run of the 5-min reaction sample, and were subjected to HPLC analysis to observe any changes. Compound **20** was spontaneously converted to DHR (**9**) after immediate administration to HPLC analysis; while the most abundant intermediate **19** was shown to be converted to **9** via **20** (Fig. 2b, traces ii, iii). A scaled up reaction of FlsO1 with **8** in 50 mM phosphate buffer (pH 6) was performed to allow the isolation of **19** for structure elucidation. A careful analysis of the 1D and 2D NMR data of **19** (Supplementary Figs. 28–34 and Tables 3 and 4) established its structure as the C-12 hydroxylated derivative of **8** (Fig. 2c), designated 12-hydroxyl-prejadomycin. Compound **20** was proposed to be a quinone compound, resulting from a spontaneous oxidation of **19** (Fig. 2c). A subsequent spontaneous dehydration of **20** gave **9**. The intermediate **21** was observed to be spontaneously transformed to **18** (Fig. 2b, trace iv), equivalent to the loss of two water molecules. Accordingly, a putative structure was proposed for **21** (Fig. 2c). Although **17** was stable in organic solvents (Supplementary Fig. 35), it was spontaneously converted to **16** under aqueous conditions (Fig. 2b, trace v). Compound **18** was also stable in organic solvents (Supplementary Fig. 36) but was spontaneously converted to **16** in aqueous solution (Fig. 2b, traces vi), with a conversation rate of 0.1849 µM/min in 50 mM phosphate buffer (pH 7.0) (Supplementary Fig. 36). The coincubation of **18** with FlsO1 did not promote the conversion efficiency (Supplementary Fig. 36). Interestingly, the faster transformation of **18** to **16** was found in 50 mM phosphate buffer with pH ≥ 7.0 (Supplementary Fig. 36). These observations demonstrated that the γ-pyrone in the xanthone moiety of **16** was probably formed through a spontaneous intramolecular Michael addition-mediated cyclization reaction of **18** and a following dehydration (Fig. 2c). Compound **16** was stable under aqueous conditions (Fig. 2b, trace vii).

A minor compound **22** was detected in the 5-min FlsO1 reaction sample by LC-MS analysis to have a molecular weight of 354 Da (Fig. 2b, trace i; Supplementary Fig. 27), which was consistent with that of the predicted epoxide-containing intermediate (Fig. 2c). When **22** was collected from an analytical HPLC assay and immediately incubated with the epoxide-capturing reagent *N,N*-diethyldithiocarbamate (DTC, **23**)[40], a compound **24** with a molecular weight of 503 Da ($m/z$ 504.3 [M + H]$^+$) was observed (Fig. 2b, trace viii; Supplementary Fig. 37), indicating the presence of an epoxide moiety in **22**. We then performed the FlsO1 reactions (30 min at 30 °C) with or without DTC (**23**). Compounds **16** and **18** were produced without adding **23** (Fig. 2b, trace ix). The presence of DTC (**23**) in the reaction completely excluded the production of **16** and **18**, in contrast, **24** was observed as the dominant product (Fig. 2b, trace x). Subsequently, a scaled up reaction of FlsO1 with PJM (**8**) and DTC (**23**) was carried out to allow the isolation of **24** (designated DTC-fluostacid A) together with an additional product **25** (designated DTC-fluostacid B). The structure determination of both compounds by NMR spectroscopic analysis (Supplementary Figs. 38–50 and Tables 3 and 4) showed the attachment of a DTC

fragment at C-6a with an opened B-ring (Fig. 2c), confirming that the parent compound **22** should have an epoxide between C-6a and C-12a. Obviously, **25** was a dehydrated derivative of **24**. We then attempted to directly characterize the structure of **22** from a large scale, short-time FlsO1 reaction with **8**. Despite exhaustive efforts, only 0.4 mg of **22** was obtained due to its inherent instability. Fortunately, the $^1$H, HMBC and HSQC NMR spectra (Supplementary Figs. 51–55 and Tables 3 and 4) clearly demonstrated the location of an epoxide moiety between C-6a and C-12a in **22**, designated epoxyprejadomycin. These cumulative data indicated that the formation of **24** likely involved a 1,3-migration of the C-12a hydroxyl to C-6 after DTC (**23**)-trapping of the epoxide at C-6a in **22**, causing a spontaneous rearrangement to open the B-ring (Supplementary Fig. 56)[41].

A minor compound **26** ($m/z$ 385.6 [M + H]$^+$) was observed in the 5-min FlsO1 reaction sample (Fig. 2b, trace i); unfortunately, it was impractical to get **26** for structure characterization due to low yield under the reaction conditions with 50 mM phosphate buffer (pH 7.0). To obtain more **26**, we optimized the FlsO1 reaction conditions by incubating FlsO1 with **8** in 50 mM phosphate buffer (pH 6.0) for a longer time (15 min) and then the reaction was terminated with methanol (MeOH), which led to an increased production of **26** and the detection of another compound **27** ($m/z$ 403.6 [M + H]$^+$) (Fig. 2b, trace xi; Supplementary Fig. 57). Next, when deuterated methanol-$d_4$ was used to stop the reaction of FlsO1, LC-MS analysis of the reaction mixture showed 3 Da increase in the molecular masses for both **26** ($m/z$ 388.6 [M + H]$^+$) and **27** ($m/z$ 406.6 [M + H]$^+$) (Supplementary Fig. 57), indicating that the incorporation of the solvent MeOH into both **26** and **27**. Subsequently, a scaled up reaction of FlsO1 and PJM (**8**, 60 mg) was carried out to allow the isolation of **26** (9.8 mg) and **27** (5.1 mg) for structure determination by NMR spectroscopic analysis (Supplementary Figs. 58–71 and Table 5). Interestingly, **27** (designated epoxyfluoxanester A) was characterized as a C-ring opened derivative with a methyl ester unit attached at C-6a and an epoxy group at C-6a/C-12a (Fig. 3a), while compound **26** (designated epoxyfluoxanester B) was a dehydrated derivative of **27**. The (6a*R*,12a*R*) absolute configuration was assigned to the epoxide in **26** and **27** by comparing the calculated and experimental ECD spectral data (Supplementary Fig. 72), and also tentatively assigned to **22**, the putative precursor of **26** and **27**.

### Mechanistical proposal of FlsO1-catalyzed xanthone formation

The characterization of putative FlsO1-reaction intermediates/shunt products leads to a proposal of the reaction mechanism for the FlsO1-catalyzed formation of **9** and xanthones **16** and **17** (Fig. 3a). The reaction is initiated by the FlsO1-catalyzed C-12 hydroxylation of **8** with the assistance of the C-4a hydroperoxy-flavin intermediate FAD-O-OH (**28**) to yield **19**, which is spontaneously oxidized to **20**, followed by a spontaneous dehydration to generate **9**. Alternatively, **20** was supposed to be the substrate for a second FlsO1-catalyzed reaction, namely the FAD-O-O$^-$ (**28**)-mediated C-6a/C-12a-epoxidation, to yield **22** (Fig. 3a). Next, FlsO1 is proposed to catalyze the third reaction, a FAD-O-O$^-$ (**28**)-assisted Baeyer-Villiger (BV) oxidation, to form the expanded 7-membered lactone ring in **29** (Fig. 3a). The intermediacy of **29** is demonstrated by characterizing epoxyfluoxanester A (**27**), a methanolyzed product of **29**, and a spontaneous dehydration of **27** gives rise to epoxyfluoxanester B (**26**) (Fig. 3a). In the FlsO1 reaction cascade, a hydrolytic opening of the lactone ring in **29** affords **30** and a subsequent decarboxylation yields the putative intermediate **31**. We proposed that **31** might diverge in two routes for xanthone formation. In route 1, a hydrolysis of the epoxide ring in **31** generates **21**. The proposed structure of **21** is inferred from the HRMS (ESI-TOF) data $m/z$ 363.1075 [M + H]$^+$ (calcd 363.1080) (Supplementary Fig. 73) and its spontaneous conversion to **18** (Fig. 2b, trace iv). A subsequent intramolecular Michael addition-mediated cyclization of **18** and a following dehydration affords the xanthone ring in **16** (Figs. 2c and 3a). In route 2, the xanthone ring in **17** is deduced to be derived from **31** by the C-7a

**Fig. 3 | Different mechanisms for the xanthone formation. a** A single FPMO-catalyzed multiple oxidations in actinomycetes. **b** A reductase and a dioxygenase-coordinated reactions in fungi. **c** A P450 enzyme-mediated C–O coupling in plant biosynthesis of xanthone.

hydroxyl-mediated epoxide ring opening (route 2). A spontaneous C-4a/C-12b dehydration of **17** with concomitant tautomerization affords **16** (Figs. 2c and 3a). Although the proposed intermediate **29** is not observed, the intermediacy of **29** is verified by the capture of the shunt products **26** and **27**. In addition, the intermediacy of **19** and **22** is confirmed by their conversions to **16** upon coincubation with FlsO1 and NADPH (Supplementary Fig. 74).

Taken together, FlsO1 is shown to catalyze three successive oxidative reactions in the conversion of **8** to **16**, including the C-12 hydroxylation (**8** to **19**), the 6a,12a-epoxidation (**20** to **22**), and the Baeyer–Villiger oxidation to insert an oxygen (**22** to **29**). Also, the epoxidation occurs prior to the Baeyer-Villiger reaction-mediated

lactone ring formation, which is supported by the presence of an epoxide in both of epoxyprejadomycin (**22**) and epoxyfluoxanester A (**27**). Since all three oxidations required $O_2$ to afford the reactive species FAD-O-O(H) (**28**), isotope-labeling experiments with $^{18}O_2$ were carried out to validate the oxygen sources in the reaction products. LC-MS analysis of the reaction intermediates/products in the presence of $^{18}O_2$ (Supplementary Fig. 75) clearly revealed the incorporation of one $^{18}O$-atom in **19** ($m/z$ 325.8 [M + H]$^+$, M + H + 2); two $^{18}O$-atoms in **16** ($m/z$ 313.8 [M + H]$^+$, M + H + 4), **17** ($m/z$ 331.8 [M + H]$^+$, M + H + 4), **22** ($m/z$ 359.7 [M + H]$^+$, M + H + 4), **24** ($m/z$ 508.3 [M + H]$^+$, M + H + 4), and **25** ($m/z$ 490.5 [M + H]$^+$, M + H + 4); and three $^{18}O$-atoms in **18** ($m/z$ 333.1 [M + H]$^+$, M + H + 6), respectively. Cumulatively, these $^{18}O$-labeling data

were fully consistent with the predicted oxidative reactions catalyzed by FlsO1 and provided further evidence to support the proposed reaction mechanism.

The FPMO XanO4 was previously shown to convert an anthraquinone precursor to the xanthone ring of xantholipin in *Streptomyces flavogriseus*, putatively via a Baeyer-Villiger oxidation, a decarboxylation, and an oxidative demethoxylation (Supplementary Fig. 76), based on the isotopic labeling result of the end product[13]; however, the mechanistic details for how XanO4 transforms an anthraquinone to a xanthone and which kind of intermediates are formed were still a black box. In this work, the actinomycete-derived FPMO FlsO1 was mechanistically characterized to be unique in catalyzing three successive oxidation steps, including a hydroxylation, an epoxidation and a Baeyer-Villiger oxidation, to afford a xanthone scaffold in two putative routes (Fig. 3a), and each step of oxidation was confirmed by identifying reaction intermediates or intermediate-related shunt products. Interestingly, the FPMO GrhO5 was recently reported by the Teufel group to catalyze the conversion of a pentangular precursor to a spiroketal moiety-containing intermediate in rubromycin biosynthesis, in which an *ortho*–hydroxylation was critical for a ring-opening step (Supplementary Fig. 77)[42,43]. Inspired by the work from the Teufel group, we realized that an *ortho*–hydroxylation by FlsO1 was an alternative mechanism to generate C-ring-opened products **30** or **27** from **22** (Supplementary Fig. 77). In this way, a C-7, C-8-β-dicarbonyl unit is formed by an *ortho*–hydroxylation at C-7a of **22**, and is ready for a retro-Claisen condensation, by which an attack of $H_2O$ (or MeOH) at the C-7 keto group would trigger the ring opening reaction to yield **30** (or **27**). However, an organic retro-Claisen condensation reaction generally requires base or Lewis acid catalysts[44,45], and would unlikely happen spontaneously. In contrast, the lactone ring in the proposed intermediate **29** is easier to encounter spontaneous hydrolysis or alcoholysis. In this work, the shunt product **27** was simply captured by adding MeOH to terminate the FlsO1 reaction, indicating that the C-ring opening reaction could happen in a mild condition. Taken together, we prefer a Baeyer-Villiger oxidation mechanism for the C–C bond cleavage in the FlsO1 reaction, and the GrhO5-like *ortho*–hydroxylation mechanism is unfavorable. It should be noted that two reactive forms of **28** (FADOOH or FADOO⁻) are proposed to perform hydroxylation (FADOOH) or epoxidation/Baeyer–Villiger oxidation (FADOO⁻) in the FlsO1-catalyzed reactions (Fig. 3a). Similarly, such two forms of reactive flavin species have been also proposed in a single FPMO enzyme-catalyzed hydroxylation/Baeyer–Villiger oxidation in the biosynthesis of legonmycin and rifamycins (Supplementary Fig. 78)[46,47].

The fungal biosynthesis of xanthones was previously proposed to require the ring opening of an anthraquinone-like precursor by a Baeyer–Villiger reaction[10,11,17], such as the transformation of chrysophanol (**32**) to monodictyphenone (**33**) (Fig. 3b). However, a very recent study reported that the ring opening reaction was actually a bienzymatic process to provide *seco*-anthraquinone (Fig. 3b), converting questin (**34**) to desmethylsulochrin (**35**) via first generating a hydroquinone intermediate (**36**) by a reductase GedF and then breaking the central ring by a dioxygenase GedK (Fig. 3b)[48]. The GedF-like reductases and GedK-like dioxygenases were conserved and coexisted in the biosynthetic pathways of diverse fungal xanthones, including demethylsterigmatocystin (AflX/AflY)[8,9], neosartorin (NsrR/NsrF)[10], agnestin A (AgnL4/AgnL3)[11], shamixanthone (MdpK/MdpL)[49,50], balanol (BlnI/BlnH)[51], cryptosporioptides (DmxR7/DmxR6)[52], and penexanthone B (PhoK/PhoJ)[53]. These data indicated a unified bienzymatic mechanism in the fungal biosynthesis of *seco*-anthraquinones, which were further modified to form the xanthone rings, such as the proposed spontaneous conversion of **37** to biennolide A (**38**) (Fig. 3b)[17], or the putative BlnE (a hypothetical protein)-catalyzed xanthone formation from **33** during balanol biosynthesis[51]. Notably, the hydroxylation of emodin anthrone (**39**) to form emodin

(**40**) in the biosynthesis of neosartorin was proposed to be catalyzed by another oxygenase NsrD (Fig. 3b)[10], which was quite similar to the FlsO1-meidated conversion of **8** to **19 → 20** (Fig. 3a). Mechanistically distinct from the microbial biosynthesis of xanthones, two P450 enzymes were demonstrate to catalyze direct C–O coupling reactions via radical species, converting a common precursor **41** to produce xanthones **42** and **43** (Fig. 3c)[16].

## Physiological function of FlsO1 implied by a substrate mimic

It was interesting to note that, in contrast to being converted to multiple products when using higher concentration of FlsO1 (such as 10 μM in Fig. 2a), PJM (**8**) was only converted to DHR (**9**) at lower concentration of FlsO1 (such as 0.5 μM) in a short time (Supplementary Fig. 79). This observation made it feasible to directly compare the kinetic parameters of FlsO1 and FlsO2 in terms of transforming **8** to **9**. Both enzymes displayed comparable $K_m$ values for **8** (FlsO1, 253.4 μM; FlsO2, 143.2 μM), and the $k_{cat}/K_m$ value of FlsO2 (0.14 min⁻¹ μM⁻¹) was only 1.8-fold greater than that of FlsO1 (0.08 min⁻¹ μM⁻¹) (Supplementary Fig. 79), which could explain the observed production of DHR (**9**) and a small amount of fluoxanthone A (**16**) in the Δ*flsO2* mutant of *M. rosaria* SCSIO N160 (Supplementary Fig. 80), for that FlsO1 should be able to act on **8** under in vivo conditions.

The putative physiological substrate **12** was not available for in vitro assays with FlsO1[32]. Alternatively, the recent isolation of NEN C (**14**)[39] provided a substrate mimic of **12**, because of their high structure similarity (Fig. 1). No consumption of **14** was observed when assaying with the control enzyme FlsO2 (Fig. 4a, traces i and ii). Intriguingly, the incubation of **14** with FlsO1 resulted in the complete consumption of **14**, but no additional products were detected (Fig. 4a, trace iii). Similar phenomena were also observed in the coupling reaction of AlpJ and AlpK with DHR (**9**), and the addition of ʟ-cysteine in the coupling reaction led to the capture of stealthin C as the product, putatively via an S-N-type Smiles rearrangement[34,35]. The addition of ʟ-cysteine in control assays yielded no new products (Fig. 4a, traces iv and v), whereas two major products **44** and **45** were detected by supplementing ʟ-cysteine into the reaction of FlsO1 with **14** (Fig. 4a, trace vi).

Next, a scaled up reaction of **14** (14 mg) and 10 μM FlsO1 in the presence of ʟ-cysteine allowed the isolation of pure **44** (3.2 mg). Unfortunately, no ¹H NMR signal of **44** was observed in a variety of deuterated solvents (Methanol-$d_4$, $H_2O$-$d_2$, and DMSO-$d_6$), which was probably due to the inherent NMR-silence property of stealthin C-related compounds[31,35]. The examples of NMR-silence compounds also included kinobscurinone and *N*-hydroxyxiamycin in kinamycin and xiamycin biosynthetic pathways, which were ascribed to co-exist with a radical species[54,55]. The UV–vis spectra of **44** and **45** were almost identical to that of stealthin C with the characteristic absorptions in the range of 440–580 nm (Supplementary Fig. 81), indicating their structure similarity. The structure of steathin C was resolved by derivatization with chemical methylations[31,35]. However, the same methylation strategy failed to provide derivatives of **44** due to the quick degradation of **44** (or its methylated products) under reaction conditions. Nonetheless, the structures of **44** and **45** (Fig. 4b) were proposed according to their high-resolution mass spectrometry (HRMS) data of **44** (*m/z* [M − H]⁻ 593.0802) and **45** (*m/z* [M − H]⁻ 577.0820) and the LC-HRMS/MS fragmentation analysis (Supplementary Fig. 82). By replacing ʟ-cysteine with ¹⁵N-labeled ʟ-cysteine in the reaction of FlsO1 and **14**, the increase of two mass units was detected for both **44** (*m/z* 595.1 [M − H]⁻, M −H + 2) and **45** (*m/z* 579.1 [M − H]⁻, M − H + 2) (Supplementary Fig. 83), confirming the presence of two ʟ-cysteine residues in **44** and **45**. When the reaction of FlsO1 and **14** was performed under ¹⁸O₂, the increase of two mass units was observed for **44** (*m/z* 595.1 [M − H]⁻, M − H + 2), but the mass unit for **45** (*m/z* 577.1 [M − H]⁻) remained unchanged (Supplementary Fig. 84). These results demonstrated that an extra oxygen from O₂ was inserted in **44**, to putatively provide the hydroxyl group at C-1 (Fig. 4b). Based on these data, we proposed that

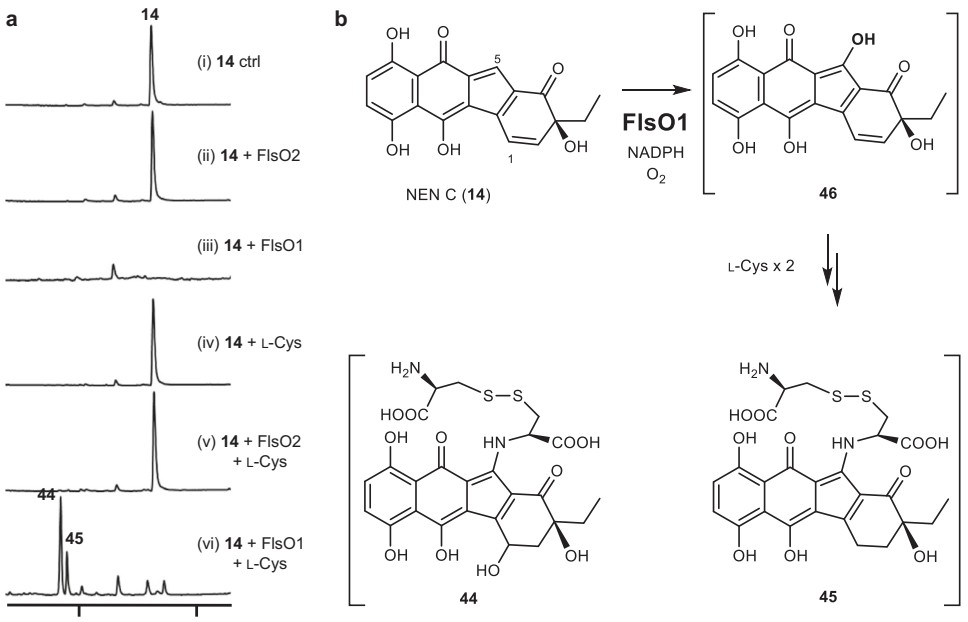

**Fig. 4 | In vitro Characterization of the physiological function of FlsO1 using NEN C (14) as a substrate mimic. a** HPLC analysis of enzyme assays. The assays were performed by incubation of 200 μM **14**, 2 mM NADPH, in the absence of L-Cys, with (i) control (no enzyme); (ii) 10 μM FlsO2; (iii) 10 μM FlsO1; or in the presence of 4 mM L-Cys, with (iv) control (no enzyme); (v) 10 μM FlsO2; (vi) 10 μM FlsO1, in 50 mM PBS buffers (pH 7.0) at 30 °C for 3 min. **b** The proposed structures of products **44** and **45** produced from **14** via the intermediate **46**.

**44** and **45** were generated from **14** in an analogous mechanism to the AlpJ/AlpK-mediated formation of stealthin C from **9** by intramolecular S−N-type Smiles rearrangement through the putative C-5 hydroxylated intermediate **46** (Fig. 4b; Supplementary Fig. 85)[35].

To our delight, the FlsO1 homologous enzymes AlpK and Nes26 from kinamycin and nenestatin parthways[34,37], respectively, also displayed similar activities as FlsO1 in converting **14** to the same products **44** and **45** by the addition of L-cysteine (Supplementary Fig. 86). These observations confirmed that FlsO1, AlpK, and Nes26 shared equivalent benzofluorene C-5 hydroxylating functions in the biosynthesis of atypical angucyclines.

## Structure analysis of FlsO1

To understand the structural basis for the substrate promiscuity and catalytic mechanism of FlsO1, the crystal structure of FlsO1 in complex with the cofactor FAD was obtained (2.3 Å, PDB ID: 7VWP; Fig. 5a; Supplementary Table 6). FlsO1 physically forms a homodimer based on the PISA calculation (Supplementary Fig. 87), similar to the homodimeric architecture of the homologous enzyme AlpK (PDB ID: 6J0Z; 2.89 Å)[56]. The electron density unambiguously delineates the binding of FAD in each FlsO1 monomer (Fig. 5a). The tertiary structure of FlsO1 (Fig. 5a; Supplementary Fig. 87) adopts a typical fold of *para*-hydroxybenzoate hydroxylase (pHBH) subfamily proteins[57], composing of the FAD binding domain (residues 4−176 and 260−374), the middle domain (residues 177−259), and the C-terminal domain (residues 374−497).

Co-crystallizing or soaking FlsO1 with the substrate **8**, in the presence or the absence of NADPH, failed to observe the density corresponding to the substrate or products. Fortunately, in the binary FlsO1 and FAD complex, the FAD conformation is close to the "in" conformation (Fig. 5a; Supplementary Fig. 87) that is suitable for the C4a peroxy-flavin intermediate FAD-O-O(H) (**28**) to deliver oxygen to the substrate, according to the well-established FAD shifting mechanism of pHBHs[57]. Therefore, this structure enables the construction of a reactive model of FlsO1 with FAD-O-O(H) (**28**), and molecular dynamic simulations were conducted to refine this model. Then we docked the natural substrate benzofluorene (**12**), the substrate mimic NEN C (**14**),

the surrogate substrate PJM (**8**), and the two proposed intermediates **20** and **22** into the FlsO1 active site. According to POCASA software[58] calculation, FlsO1 exhibits a broad substrate binding cavity (355 Å³, Fig. 5a). All five compounds can be effectively sealed into the substrate cavity, and be properly positioned for the reactive C4a-peroxyl of FAD-O-O(H) (**28**) to deliver oxygen to the proposed sites (Fig. 5b−f). Notably, compounds **8**, **20**, and **22** are similarly orientated in the active pocket, suggesting that the unusual stepwise oxidations on **8**, **20**, and **22** only require slight swings of the intermediates in the active site of FlsO1 (Supplementary Fig. 88).

Roles of key substrate binding residues (H76, R95, M203, L205, V215, F258, and M284) of FlsO1 were probed by the approach of site-directed mutagenesis, using **8** as the substrate (Fig. 5g; Supplementary Fig. 89). The mutants H76F, R95A, V215G, F258A, and M284A displayed decreased activities, judging from their ability by consuming less **8** than the wild type FlsO1, which may result from the weakened substrate binding. Interestingly, the M203A mutant completely lost the catalytic activity, whereas the mutant L205A showed a slight increase in the catalytic efficiency. Analysis of the docked model shows that the conjugated B–D ring systems of the substrate **8** is sandwiched between the loop G287G288Q289 and the side chains of M203 and L205 (Fig. 5h), and the ring A further stacks with L205. We propose that the M203A significantly alters the orientation of **8**, thereby losing the catalytic activity; whereas the L205A releases the steric clash with the ring A and promotes the reaction activity. We also probed the FAD binding residues R45 and R213. The resultant mutants R45A and R213A indeed completely lost the catalytic activity.

Phylogenetic analysis reveals that FlsO1 is well clustered with type II group A FPMOs and thus is separated from the GrhO5-type I enzymes[43] (Supplementary Fig. 90). To further explore why FlsO1 recognizes both PJM (**8**) and NEN C (**14**), and catalyzes multistep oxidative reactions on PJM (**8**), structure comparison was made for type II group A FPMOs FlsO1, AlpK[56] that only recognizes **14**, PgaE and CabE that perform the C-12 hydroxylation on **8**[59,60]. Amino acid sequence alignment analysis shows that FlsO1 shares moderate sequence identity with AlpK (63.1%), PgaE (53%) and CabE (56%). The crystal structures of these FPMOs are overall conserved. However,

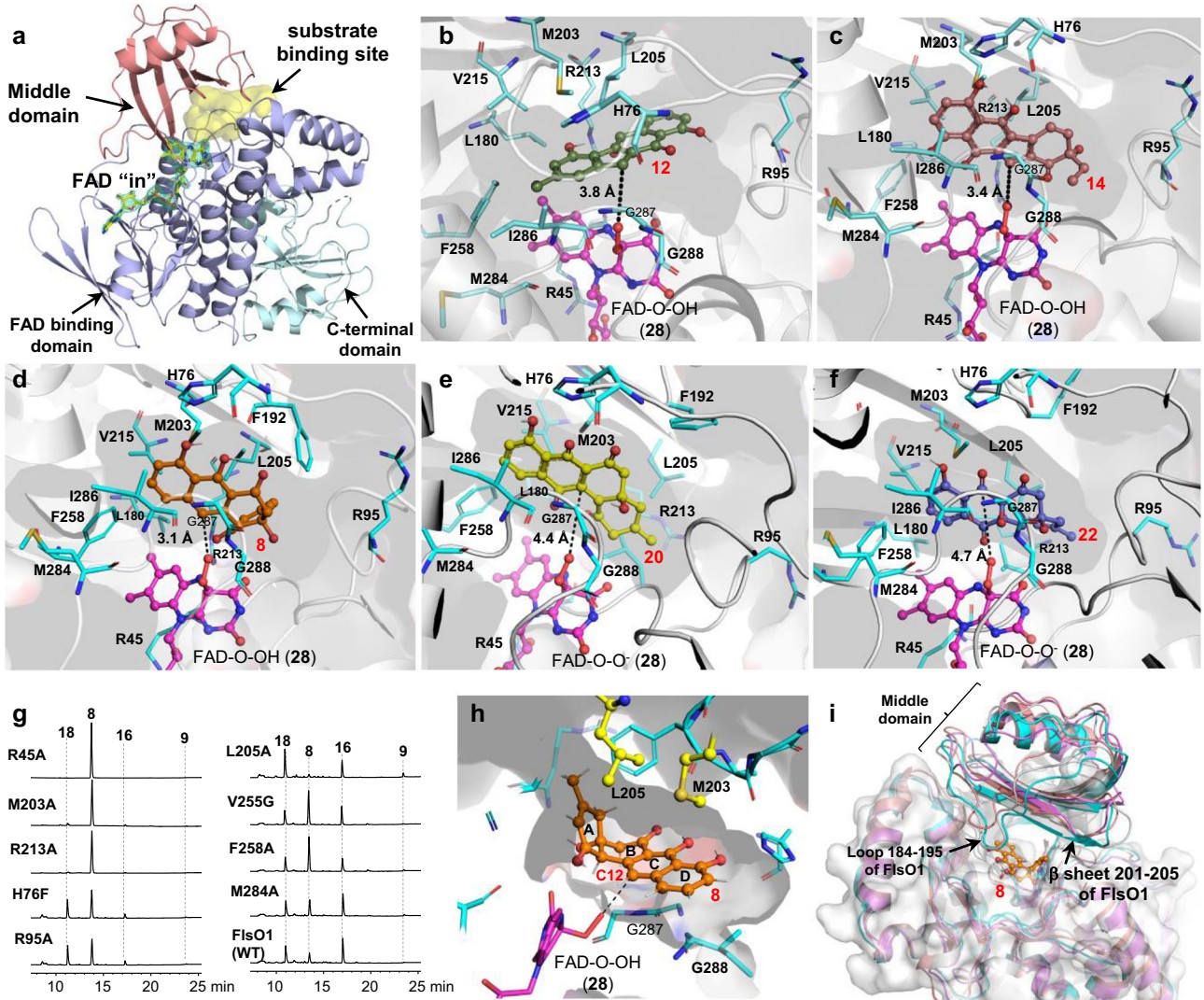

**Fig. 5 | Structural analysis of FlsO1. a** Crystal structure of FlsO1 in complex with FAD. The $2F_O$-$F_C$ map (contoured at 2.5 σ) of FAD is shown as green mesh. Note that FAD adopt an "in" conformation. The substrate binding pocket predicted by POCASA is shown in yellow surface. Predicted binding modes of (**b**) the native substrate benzofluorene (**12**); **c** NEN C (**14**); **d** PJM (**8**); and the two proposed intermediates (**e** **20** and (**f**) **22**. The compounds are shown as colored sticks and spheres for **12** (smudge), **14** (salmon), **8** (orange), **20** (yellow) and **22** (lightblue), respectively. Notably, the intermediate **22** adopts similar orientation as **8** and **20**, despite a horizontal shift. The active site residues are shown as sticks. The distance between the oxygen of FAD-O-OH (**28**) and the proposed oxidized site of the substrates are measured (black dashes) and labeled. **g** HPLC analysis of reactions of FlsO1 or its mutants with PJM (**8**). **h** Highlighted interactions of M203 and L205 with PJM (**8**). **i** AlpK (violet cartoon), PgaE (gray) and CabE (simmon) aligned with FlsO1 (cyan). To indicate the substrate binding site, PJM (**8**) docked in FlsO1 is shown as gold sticks. The middle domain is not well aligned, especially for the loop 184−195 and β sheet 201−205 of FlsO1 that construct the substrate binding pocket.

the middle domain relating to substrate binding and release are not well aligned. Especially, the deduced substrate entrance/product exit region is not conserved or adopts highly distinct conformations in FlsO1 (loop 184−195 and β sheet 201−205), AlpK (loop 185−196 and 202−206), PgaE (181−192 and 198−202) and CabE (181−192 and 198−202) (Fig. 5i), which may account for their substrate selectivity and reactivity. The generation of the inactive mutant M203A and the activity-enhanced mutant L205A by mutagenesis studies of FlsO1 confirms the importance of residues in the region of substrate entrance/product exit.

In conclusion, this work provides biochemical evidence to support the physiological function of FlsO1 (as well as its homologous enzymes AlpK and Nes26) as a benzofluorene C-5 hydrxoylase in the biosynthesis of atypical angucyclines by using the natural substrate mimic NEN C (**14**). In addition, FlsO1 is demonstrated to be a versatile and multifunctional FPMO to convert PJM (**8**) to multiple products including DHR (**9**), the xanthone products **16** and **17**, and the C-ring

opened product **18**. The underlying mechanism is revealed by detailed characterization of reaction intermediates/shunt products (such as **19**, **22**, **24−27**) to undergo multistep oxidations of hydroxylation, epoxidation and Baeyer-Villiger oxidation. The substrate flexibility and functional diversity of FlsO1 can be explained by a broad substrate binding cavity in the resolved crystal structure of FlsO1. This study highlights a single FPMO-mediated epoxidation/BV oxidation strategy in providing *seco*-angucyclinone for the xanthone formation in actinomycetes, which is distinct from the bienzymatic (reductase/dioxygenase) strategy in fungi and the P450 enzyme-mediated radical C−O coupling strategy in plants.

## Methods

### Bacterial strains, plasmids, and reagents
Primers, bacterial strains, and plasmids used and constructed in this study are listed in Supplementary Tables 1 and 2. Chemicals, enzymes, and other molecular biological reagents were purchased from

standard commercial sources and used according to the manufacturers' recommendations.

## DNA isolation, manipulation, and sequencing

DNA isolation and manipulation in *Escherichia coli*, *M. rosaria* SCSIO N160[21], *M. echinospora* SCSIO 04089[37] and *Streptomyces ambofaciens* Δ*alp1U*[61] were carried out according to standard procedures. Primers were synthesized at The Beijing Genomics Institute (BGI). PCR amplifications were carried out on an Authorized Thermal Cycler (Eppendorf AG). DNA sequencing was performed at the iGeneTechTM Biotech Co., Ltd. (Guangzhou) and TsingKe Biological Technology Co. (Guangzhou).

## General experimental procedures

$^1$H and $^{13}$C NMR spectra were recorded on either a Bruker Avance Bruker 700 spectrometer with tetramethylsilane (TMS) as the internal standard. High-resolution electro-spray ionization mass spectrometric (HRESIMS) data were measured on a MaXis 4G UHR-TOFMS spectrometer (Bruker Daltonics Inc.). Column chromatography (CC) was performed using silica gel (100–200 mesh, Jiangyou Silica Gel Development, Inc., Yantai, P. R. China) and Sephadex LH-20 (GE Healthcare Bio-Sciences AB, Sweden). Medium-pressure liquid chromatography (MPLC) was performed on an automatic flash chromatography system (CHEETAHTM MP 200, Bonna-Agela Technologies Co., Ltd., China) with the monitoring wavelength at 304 nm and the collecting wavelength at 254 nm. Semipreparative HPLC was performed on a Hitachi HPLC (Hitachi- L2130, Tokyo, Japan) with a Diode Array Detector (Hitachi L-2455) using a Phenomenex ODS column (250 mm × 10.0 mm, 5 μm; Phenomenex, USA). Preparative thin layer chromatography (TLC, 0.1–0.2 or 0.3–0.4 mm) was conducted with precoated silica gel GF254 (10–40 nm, Yantai) glass plates. Assignments of NMR data were based on DEPT, HSQC, COSY, HMBC, and NOESY experiments.

## Expression and purification of FlsO1, FlsO2, FlsO3, FlsO4, FlsO5, AlpK, and Nes26

DNA fragments carrying the target genes including *flsO1* and its homologs *flsO2*, *flsO3*, *flsO4,* and *flsO5* (GenBank accession number KT726162.1 [https://www.ncbi.nlm.nih.gov/nuccore/KT726162.1]) were PCR amplified from the genomic DNA of *M. rosaria* SCSIO N160[21]. The DNA fragment encoding AlpK (GenBank accession number AY338477.2 [https://www.ncbi.nlm.nih.gov/nuccore/AY338477.2]) and Nes26 (GenBank accession number KY454837.1 [https://www.ncbi.nlm.nih.gov/nuccore/KY454837.1]) were amplified from the genomic DNA of *S. ambofaciens* Δ*alp1U*[61] and *M. echinospora* SCSIO 04089[37], respectively. The purified PCR products were subcloned into pET28a to yield the plasmids pCSG5205 (for *flsO1* expression), pCSG5203 (for *flsO3* expression), pCSG5210 (for *flsO4* expression), pCSG5215 (for *flsO5* expression), pCSG5230 (for *alpK* expression) and pCSG5231 (for *nes26* expression) (Supplementary Tables 1 and 2). The protein expression vector pCSG5102 for *flsO2* expression was previously constructed[28]. The overexpression of *flsO1* was carried out in *E. coli* BL21(DE3)/pCSG5205 in LB media containing 50 μg/mL kanamycin at 16 °C for 20 h by the induction of 0.1 mM IPTG (isopropylthio-β-galactoside). Purification of (His)$_6$-tagged FlsO1 was conducted using Ni-NTA affinity chromatography according to the manufacturer's manual (Novagen, USA). The purified FlsO1 was desalted with PD-10 column (GE Healthcare, USA) and stored in the storage buffer (10% glycerol, 1 mM DTT, 50 mM Tris-Cl, 100 mM NaCl, pH 8.0) at −80 °C. The recombinant proteins FlsO2, FlsO3, FlsO4, FlsO5, AlpK, and Nes26 were prepared using similar methods. The concentration of purified enzymes was determined using a NanoDrop UV-Vis spectrophotometer (Thermo Fisher Scientific).

## In vitro enzyme assays

A typical in vitro enzyme reaction of FlsO1 (or its homologous enzymes) was conducted in 100 μL phosphate buffer (50 mM, pH 7.0) comprising of 200 μM PJM (**8**), 2 mM NADPH, 10 μM FlsO1 (or FlsO2, FlsO3, FlsO4, FlsO5, AlpK, and Nes26). The reactions were incubated for 30 min at 30 °C and then were quenched by mixing with 100 μL of ice-cold MeOH. When using NEN C (**14**) as the mimic substrate, the reaction was conducted in a 100 μL reaction mixture consisting of 200 μM **14**, 10 μM FlsO1 (or AlpK, Nes26), 2 mM NADPH with or without 4 mM L-cysteine in 50 mM phosphate buffer (pH 7.0). The reaction mixtures were incubated for 30 min at 30 °C and were stopped by adding 100 μL ice-cold MeOH. HPLC analysis of the enzyme reactions was carried out on the Agilent 1260 Infinity series instrument (Agilent Technologies Inc., USA) using a reversed phase column Luna C18 (5 μm, 150 × 4.6 mm, Phenomenex) with UV detection at 304 nm under the following program: solvent system (solvent A, 10% MeCN in water supplemented with 0.1% formic acid; solvent B, 90% MeCN in water); 5% B to 80 % B (0–20 min), 100% B (21–24 min), 100% B to 5% B (24–25 min), 5% B (25–30 min); flow rate at 1 mL/min. For the analysis of reaction with NEN C (**14**), 0.1% formic acid in solvent A was replaced by 0.1% TFA (trifluoroacetic acid).

## Determination of kinetic parameters of FlsO1 and FlsO2 toward 8

For determining the kinetic parameters of FlsO1-catalyzed reaction, PJM (**8**) was set at the concentrations of 15, 25, 50, 75, 100, 150, 200, 250, 300, 400, 1000, and 1500 μM. Enzyme assays were performed in triplicates in 50 mM phosphate buffer (pH 7.0) with 0.5 μM FlsO1 and 2 mM NADPH, by incubation at 30 °C for 4 min. For determining the kinetic parameters of FlsO2-catalyzed reaction, PJM (**8**) was set at the concentrations of 15, 25, 50, 75, 100, 150, 250, 500, and 1000 μM. Enzyme assays were performed in triplicates in 50 mM phosphate buffer (pH 7.0) with 0.25 μM FlsO2 and 2 mM NADPH, by incubation at 30 °C for 6 min. HPLC was used to analyze the enzyme reactions of FlsO1 and FlsO2, with detection wavelength at 304 nm. For both FlsO1 and FlsO2, the relative yields of DHR (**9**) were used to calculate the conversation rate and velocity. Kinetic parameters ($K_m$, $k_{cat}$, $V_{max}$) were determined by nonlinear regression analysis using the GraphPad Prism 6 software.

## $^{18}$O$_2$ labeling experiments

For the $^{18}$O-labeling reaction with PJM (**8**), a stream of high-purity nitrogen was bubbled into a 50 μL FlsO1 reaction system (containing 200 μM **8**, 10 μM FlsO1, 2 mM NADPH in 50 mM phosphate buffer, pH 6.0) for 5 min to thoroughly remove the atmospheric O$_2$. Subsequently, the 97% $^{18}$O$_2$ gas (Shanghai research institute of chemical industry Co. Ltd.) was bubbled into the reaction system for 3 min. The process of purging air O$_2$ by nitrogen gas and filling the reaction system with $^{18}$O$_2$ was kept and conducted on the ice. Afterwards, the reaction mixtures were centrifuged and incubated at 30 °C for 35 min and then quenched by adding 50 μL of ice-cold MeOH. The reaction mixtures were then subjected to LC-MS analysis. A similar assay with $^{16}$O$_2$ in atmosphere environment was used as the control. For the $^{18}$O-labeling reaction with NEN C (**14**), a stream of nitrogen was bubbled into the FlsO1 reaction system comprising 200 μM **14**, 10 μM FlsO1, 2 mM NADPH and 4 mM L-cysteine in 50 μL of 50 mM phosphate buffer (pH 7.0) for 5 min to thoroughly remove the atmospheric O$_2$. Subsequently, the 97% $^{18}$O$_2$ gas was bubbled into the reaction system for 3 min. The entire process was also conducted on the rice. Afterwards, the reaction mixtures were centrifuged and incubated at 30 °C for 1–3 min and then quenched by adding 50 μL of ice-cold MeOH for subsequent LC-MS analysis. A similar assay with $^{16}$O$_2$ in atmosphere environment was used as the control.

## $^{15}$N-L-cysteine labeling experiments

The $^{15}$N-L-cysteine labeling assays were conducted in 50 μL of 50 mM phosphate buffer (pH 7.0) consisting of 10 μM FlsO1, 200 μM NEN C (**14**), 2 mM NADPH and 4 mM $^{15}$N-L-cysteine. Upon incubation at 30 °C for 1–3 min, the reaction was stopped by mixing with 50 μL of ice-cold MeOH, followed by centrifugation at 4 °C, 7200 × $g$ for 10 min. The supernatants were transferred to a new 1.5 mL microcentrifuge tube for further HPLC and LC-HRMS analysis. A similar reaction with L-cysteine was performed as the control.

## Enzymatic preparation of 16–19 and 22

For the preparation of fluoxanthone A (**16**), fluoxanthone B (**17**) and fluoxanol (**18**), the FlsO1 (10 μM) reaction was carried out with 52 mg PJM (**8**) and 2 mM NADPH in 1.6 L of 50 mM phosphate buffer (pH 7) by incubation at 30 °C for 45 min and then extracted three times with an equal volume of EtOAc. Subsequently, the organic phase was collected and evaporated to dryness, and then redissolved in 4.5 mL MeOH. Compounds **16–18** were purified by semi-preparative HPLC with biphasic solvents (solvent A: 10% MeCN in water supplemented with 0.1% formic acid, solvent B: 90% MeCN in water). Finally, purification upon the semi-preparative HPLC yielded 6.0 mg **16**, 2.6 mg **17,** and 12.3 mg **18**. For the preparation of **19** and **22**, the reaction was prepared with 10 μM FlsO1, 30 mg **8** and 2 mM NADPH in 50 mM phosphate buffer (pH 6) upon incubation at 30 °C for 4 min and then extracted three times with an equal volume of EtOAc. Subsequently, the organic phase was collected and evaporated to dryness, and then redissolved in 3 mL MeOH. Compounds **19** and **22** were purified by semi-preparative HPLC using biphasic solvents (solvent A: 10% MeCN in water supplemented with 0.1% formic acid, solvent B: 90% MeCN in water). Since **19** and **22** were highly instable, both compounds (16.0 mg **19** and 0.4 mg **22**) were afforded by freeze-drying at −40 °C.

## Trapping of epoxide with *N,N*-diethyldithiocarbamate (DTC, 23) and enzymatic preparation of 24 and 25

To trap putative epoxide intermediates, the reaction was conducted in 50 μL of 50 mM phosphate buffer (pH 7.0) comprising of 10 μM FlsO1, 200 μM PJM (**8**), 1 mM NADPH, 5 mM sodium DTC (**23**). The mixtures were incubated at 30 °C for 6 min and then the reaction was terminated by mixing with 50 μL of ice-cold MeOH. After centrifugation, the supernatants were subjected to HPLC and LC-HRMS analysis.

To isolate the product **24**, the FlsO1 reaction was proportionally scaled up to a 300 mL volume. After incubation at 30 °C for 6 min, the reaction was terminated by adding 300 mL of ice-cold MeOH. The reaction mixture was freeze-dried and then extracted five times by MeOH. The extracts were redissolved in 4.5 mL MeOH. Finally, semi-preparative HPLC afforded 8.5 mg **24** and 4.3 mg **25**.

## Enzymatic Preparation of 26 and 27

For the preparation of **26** and **27**, the enzymatic reaction was prepared with 10 μM FlsO1, 60 mg **8**, and 1.5 mM NADPH in 50 mM phosphate buffer (pH 6.0) upon incubation at 30 °C for 15 min. The reaction was then terminated by mixing with equal volume of ice-cold MeOH. After evaporating methanol, the aqueous phase was extracted three times, each with an equal volume of EtOAc. Next, the organic phase was collected and evaporated to dryness, and then redissolved in 4 mL MeOH. Compounds **26** and **27** were purified by semi-preparative HPLC using biphasic solvents (solvent A: 10% MeCN in water supplemented with 0.1% formic acid, solvent B: 90% MeCN in water). Finally, compounds **26** (9.8 mg) and **27** (5.1 mg) were afforded by freeze-drying at −40 °C.

## Determination of the nonenzymatic conversion rate of 18 to 16

The conversion of **18** to **16** was performed by incubation of 100 μM **18** in different solvents (ultra-pure water, MeOH, DMSO, acetone, EtOAc and CHCl$_3$) overnight at 30 °C. The pH-dependent stability of **18** was assayed by incubation of 100 μM **18** at 30 °C for 2 h in 50 μL of 50 mM phosphate buffer saline (PBS) or 50 mM citric acid/ sodium citrate buffer with pH values ranging from pH 4.0 to pH 9.0. To test the conversion rate of nonenzymatic conversion of **18** to **16**, a time course assay was conducted by incubation of 100 μM **18** at 30 °C in 50 mM phosphate buffer (pH 7.0), and samples were taken at 1 h, 2 h, 3 h, 4 h, 6 h, and 9 h. The assays were done in triplicates. The curve of the incubation time versus the remaining concentrations of **18** was obtained by comparing with the concentration of fluostatin C (**7**) as an internal standard.

## Preparation of compounds 44 and 45 from 14

The reaction was carried out with 5 μM FlsO1 and 14 mg NEN C (**14**) in 200 mL of 50 mM phosphate buffer (pH7.0) containing 1 mM NADPH and 4 mM L-cysteine and was incubated at 30 °C for 15 min to prepare **44** and **45**. The reaction was stopped by adding 200 mL ice-cold MeOH. Since **44** and **45** could not be extracted by normal organic solvents, the reaction mixtures were concentrated by freeze-drying. The resulting sediment was then redissolved in 10 mL ultra-pure water and subjected to semi-preparative HPLC. Finally, **44** (3.2 mg) and **45** (0.6 mg) were obtained and subjected to NMR analysis in a variety of deuterated solvents (MeOH-$d_4$, H$_2$O-$d_2$, and DMSO-$d_6$) to reveal the inherent NMR silence property of both **44** and **45**.

## Construction and overexpression of site-specific mutation of FlsO1

Site-directed mutagenesis was carried out according to manufacturer's instructions (TransGen) to generate FlsO1 mutants R45A, H76F, R95A, M203A, L205A, R213A, V215G, F258A, M284A, and Q108G/R109G. The DNA fragments carrying the mutated sites were amplified with primers listed in Supplementary Table 2 by using plasmid pCSG5205 as the template. The individual mutations in the constructs were confirmed by sequencing. The mutant plasmids were then expressed in *E. coli* BL21(DE3) and the corresponding proteins were purified as described above.

## Protein crystallization and structural elucidation of FlsO1

For crystallization, the target protein FlsO1 is further purified using superdex 200 in 20 mM Tris buffer (pH 7.5) containing 200 mM NaCl and 1 mM DTT, and was concentrated to ~20 mg mL$^{-1}$ using a Ultracel 30 K filter (Millipore). Bright yellow crystals of FlsO1 can grow from the wild-type FlsO1 proteins, which were then diffracted at the Shanghai Synchrotron Radiation Facility. Data reduction and integration were processed with HKL2000 package or XDS. However, despite extensive optimization efforts, the crystals of wild-type FlsO1 can only diffract to a limit of 2.89 Å resolution. To improve the diffraction quality, the double mutation of two surface residues of FlsO1 (Q108G/R109G) was designed by the Surface Entropy Reduction prediction (SERp) server[62]. The Q108G/R109G mutant was confirmed to have the same activity and product profiles as the wild-type FlsO1 (Supplementary Fig. 91). This mutant was further screened with the crystallization conditions, in the presence of NADP$^+$ and NADPH, respectively. Cube-like crystals appeared in one week, in the condition of PACT premier™ kit of Molecular Dimensions (0.02 M Sodium/potassium phosphate, 0.1 M Bis-Tris propane, pH 6.5, 20% w/v PEG 3350). The crystal can diffract to 2.3 Å, using the in-house Rigaku XtaLAB Pro: kappa single device equipped with rotating anode X-ray source (λCu Kα = 1.54184 Å) and Pilatus 3R 200K-A detector and processed using CrysAlisPro version 1.0.39. The initial structure model was built by the molecular replacement method using PHENIX with the PgaE structure (PDB ID: 2QA1[60] [https://doi.org/10.2210/pdb2QA1/pdb]) as the molecular replacement search model and then refined using REFMAC. COOT was used for model adjustments. The qualities of the final model were validated by MolProbity. Structural diagrams were prepared using the program PyMOL (http://www.pymol.org/).

## Substrates docking and MD simulations studies of FlsO1

Using the Chain D of FlsO1/FAD structure as the receptor, PJM (**8**) was first docked using Autodock vina[63]. The geometrical restraints for **8** and the following ligands are generated by Grade Web Server (http://grade.globalphasing.org). The resulting docked model, with FAD manually modified to FAD-O-O(H) (**28**), was conducted with 100 ns molecular dynamics (MD) simulation study. From the resulted MD trajectories, one trajectory profile with the FAD-O-O(H) (**28**) in traditional "in" conformation was selected for further docking analysis. In details, MD simulations were performed using the Desmond package of Schrödinger software (Schrodinger LLC. 2020). Geometric chemical structure corrections in complexes were performed using Protein Preparation Wizard module, including replenishing the missing hydrogen atoms and side chains; $p$Ka was determined by propka and Epik for protein and ligand, respectively; restrained minimization (RMSD < 0.3 Å for heavy atoms) was performed using OPLS3e force field. The molecular system was solvated with water (TIP3P) molecules under orthorhombic periodic boundary conditions for a 10 Å buffer region and the system was neutralized by adding $Na^+$ as counter ions. A 100 ps energy minimization simulation was performed on the system using Brownian motion simulation. In the process of pre-equilibration step, the NVT simulation was introduced for 5 ns; the force constants of 10 and 5 kcal/mol/Å$^2$ are respectively imposed on the backbone and side chains of the system; then the side chain was fully relaxed with NPT ensemble for 5 ns with position constraint for the backbone (5 kcal/mol/Å$^2$). Subsequently, for the production running step, the MD simulation process of 100 ns was performed with NPT ensemble. MD trajectories were recorded every 20 ps intervals. The key atomic pair interaction was analyzed with VMD software[64]. For further docking of FlsO1/FAD-O-O(H) (**28**) with **8**, **12**, **14**, **20**, and **22**, flexible docking was applied, i.e., residues L47, R95, F192, L205, and R213 in substrate binding site were allowed to rotate. A grid box of a 22.5 Å × 21 Å × 18.75 Å size was centered on the catalytic site.

## Reporting summary

Further information on research design is available in the Nature Research Reporting Summary linked to this article.

## Data availability

Data generated in this study are available within the paper and its Supplementary information files. The GenBank accession number of *fls* genes (*flsO1*, *flsO2*, *flsO3*, *flsO4* and *flsO5*) is KT726162.1 [https://www.ncbi.nlm.nih.gov/nuccore/KT726162.1]. The GenBank accession number of *alpK* is AY338477.2 [https://www.ncbi.nlm.nih.gov/nuccore/AY338477.2]. The GenBank accession number of *nes26* is KY454837.1 [https://www.ncbi.nlm.nih.gov/nuccore/KY454837.1]. Crystallographic data for FlsO1 were deposited in the Protein Data Bank (PDB) with accession codes 7VWP [https://doi.org/10.2210/pdb7VWP/pdb]. The structures were obtained from the Protein Data Bank (PDB) with accession codes 6J0Z (AlpK) [https://doi.org/10.2210/pdb6J0Z/pdb], 2QA1 (PgaE) [https://doi.org/10.2210/pdb2QA1/pdb], 2QA2 (CabE) [https://doi.org/10.2210/pdb2QA2/pdb], 4K5S (MtmOIV) [https://doi.org/10.2210/pdb4K5S/pdb], and 4X4J (BexE) [https://doi.org/10.2210/pdb4X4J/pdb]. Source data are provided with this paper. Data is available from the corresponding authors upon request. Data for this manuscript are also available at South China Sea Ocean Data Center, National Earth System Science Data Center, National Science & Technology Infrastructure of China [http://data.scsio.ac.cn/metaData-detail/1563148765659291648].

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

## Acknowledgements

This work is supported in part by the National Natural Science Foundation of China (31820103003 to C.Z., 31700042 to C.Y., 22193072 to L.Z., 42176127 to W.Z., and 41676165 to W.Z.); Key Science and Technology Project of Hainan Province (ZDKJ202018 to C.Z.); Chinese Academy of Sciences (QYZDJ-SSW-DQC004 to C.Z.); MOST (2018YFA0901903 to Y.Z.); K.C. Wong Education Foundation (GJTD-2020-12 to C.Z.); the Guangdong Provincial Special Fund for Marine Economic Development Project (GDNRC[2021]48 to C.Z.); Southern Marine Science and Engineering Guangdong Laboratory (Guangzhou) (GML2019ZD0406 to C.Z.); Youth Innovation Promotion Association CAS (2022349 to C.Y.) and the Science and Technology Planning Project of Guangzhou (202102020471 to C.Y.). We appreciate Prof. Keqiang Fan in Institute of Microbiology, Chinese Academy of Sciences for generous gift of *Streptomyces ambofaciens Δalp1U*. We are grateful to Dr. Z. H. Xiao, X. H. Zheng, A. J. Sun, Y. Zhang and X. Ma in the equipment public service center at SCSIO for recording spectroscopic data, and thank Dr. X. M. Xu in Ocean University of China for help in molecular dynamics (MD) simulation studies. We also thank the data archive support from the National Earth System Data Center, National Science & Technology Infrastructure of China (http://www.geodata.cn).

## Author contributions

C.Y., W.Z., C.H., X.J., and W.L. performed compound isolation and structure determination. C.Y., Y.Z., M.Z., and B.C.D., conducted the in vitro biochemical studies and analysis. L.Z. carried out the protein crystallization experiments, X-ray analysis, the crystal structure determinations of the proteins, and molecular docking analysis. C.Y., L.Z. W.Z., and C.Z. analyzed the data and wrote the manuscript. C.Z. directed the research. C.Y. and L.Z. contributed equally to this work.

## Competing interests

The authors declare no competing interests.
