## [Peer Review File · Nature Communications]

Biochemical and structural insights of multifunctional flavin-dependent monooxygenase FlsO1-catalyzed unexpected xanthone formationREVIEWER COMMENTS

Reviewer #1 (Remarks to the Author):

The manuscript of Yang et al. provides new insights in the way a single monooxygenase is able to form the xanthone moiety in bacterial secondary metabolites. Except for thorough chemical analyses, also details on the enzyme structure and mechanism are reported. The study has been carefully performed, and contains many interesting data.

While the results are novel, there is already quite some information on related flavoprotein monooxygenases, particularly those involved in microbial secondary metabolite pathways. Especially, the work performed on the XanO4 contains already leads for how such enzyme can form a xanthone. In that context, this manuscript is somewhat confirmative, building on previous results and hypotheses.

Also, the observed products and intermediates still cannot rule out alternative mechanisms. The proposed scheme in Figure 3a contains several intermediates that have not been experimentally verified. While the data are convincing for the first two oxygenations, the third seems a bit out of place; it would involve a Baeyer-Villiger oxidation. This is at odds with the first two steps that would rely on a hydroperoxyflavin intermediate, the BV oxidation is supposed to be catalyzed by a peroxyflavin intermediate. No explanation is given how/why the enzyme would be able to form two different enzyme intermediates. In fact, one could imagine that also the third oxygen insertion is a hydroxylation, but then ortho, instead of para in the first hydroxylation (installing an OH 'left' from the carbon that will be removed). Such highly hydroxylated compound can also result in lactone formation. In this context, it is a bit surprising that a recent paper by the Teufel group was not included in the discussion (doi: 10.1002/anie.202109384) where sequence-related bacterial monooxygenases have been reported, including a structure with substrate ligands. Also in that study, hydroxylation leads to ring opening, without the need for a B-V reaction.

Some text is a bit difficult to follow. For example (in the abstract), in the abstract the terms 'xanthone framework', 'the xanthone formation' and 'confirmative evidence' should be better introduced/explained. Also, FMO is not the proper term to use as it is commonly used for a subclass of flavin-containing monooxygenases that are not related to the reported enzymes. Please use FPMOs (flavoprotein monooxygenases) to avoid confusion (original ref for classification: doi: 10.1016/j.jbiotec.2006.03.044).

Be careful with providing original/proper references: when discussing the PHBH family and in/out position of the FAD cofactor, refs 48&49 are not the most logical, consider doi: 10.1096/fasebj.9.7.7737455.

The remark that the group of FPMOs studies in this manuscript is in some way sequence-related to two-component BVMOs and fungal cofactor-independent dioxygenases is totally wrong. The respective Fig S68 should be removed. It would be better to have a complete as possible tree of sequences closely related to FlsO1.

Reviewer #2 (Remarks to the Author):

The authors presented an interesting study about the versatile flavin-dependent monooxygenase FlsO1. Several different oxidation reactions can be catalyzed by FlsO1, including hydroxylation, epoxidation and Baeyer-Villiger oxidation. Substrate mimic, isotope labeling, crystal structure of FlsO1 and MD simulations were performed well to confirm the biocatalytic mechanisms. I recommend this paper to be published in Nature Commun. after major revisions. Several comments are listed to improve the quality of the paper for the possible publication in Nature Commun.

Major points:

In Fig. 5 def, the orientation of intermediate 22 in the active pocket was different from 8 and 20. FadOOH was closed to C12 in the hydroxylation of compound 8, but to C7 in the Baeyer-Villiger oxidation of 22. According to the structure of 22, two carbonyls (C12 and C7) could be the reactive

sites, so the distance of C12 and fadOOH should be given in Fig. 5f. The two reactive sites could lead to the different xanthones in structure. Indeed, it is difficult to distinguish the position of oxygen atom in xanthone skeleton. I suggest a crystal structure of 16, 17 or 18 should be added.

Minor points:

- (1) In line 157 and Supplementary Fig. 55, the authors claimed a 1,3-migration in the reaction of 22 to 24. I noticed this process was hypothesized to be catalyzed by H⁺, but performed under neutral condition indeed (Fig. 2c). Ref. 39 reported a rearrangement of N-Acyl Isothioureas. Considering the difference between imine and hydroxyl, I suggest more experiments or references to disclose this process.
- (2) The compound 16 should be further purified, according to its ¹HNMR and ¹³CNMR (Fig. 5, 6 in SI).
- (3) I notice that the isotope incorporation in ¹⁸O₂ labeling experiments is significantly lower than 97% (the gas be used), I am curious why it decreased.
- (4) The format of references should be unified.

Reviewer #3 (Remarks to the Author):

The authors report the functional and structural characterization of a flavin-dependent oxygenase FlsO1. FlsO1 is demonstrated *in vitro* to be able to complement another oxygenase FlsO2 by converting pre-jadamycin B to dehydrorebelomycin b, but also convert pre-jadamycin to two xanthone products. FlsO1's "natural activity" for hydroxylating a benzo[b]-fluorene intermediate was also demonstrated using a surrogate substrate called nenestain C. A substantial amount of data is provided, and structural assignments of enzyme products are assigned by extensive spectroscopic analysis. The manuscript, although dense, is well written and well organized. The results are interesting and will be valuable to the community.

In general the conclusions are supported by the data. The exception is the statement that FlsO1 catalyzes cryptic xanthone formation. In the biosynthetic field, cryptic is often used to describe an unexpected intermediate in a biosynthetic pathway, but the xanthones produced here are apparently the final product. More importantly, it's not clear whether the xanthones have any significance as natural products from this strain. Are xanthones detected from *Micromonospora rosaria* SCSIO N160 or the flsO2-inactivation mutant? It's most likely that xanthone formation is a consequence of *in vitro* conditions wherein the incorrect substrate is provided to the enzyme. This is supported by the kinetic data provided by the authors, which suggests the conversion to xanthones is not efficient and simply an off-pathway reaction that is a consequence of *in vitro* characterization. It would be useful to note if this mechanism of xanthone formation is actually expected to naturally occurring.

Regarding the kinetic data collected with FlsO1 and FlsO2 with compound 8: what is the product that is being detected to calculate the velocity? Based on the HPLC trace in Figure 2a at pH 7, it appears that very little 9 is being formed, while 16 and 18 are relatively abundant. Is the calculated velocity through detecting 9? How does the kinetic analysis with respect to 16 and 18 compare? Similarly, the HPLC analysis of the FlsO2 shows that 10 is formed in greater amounts than 9. Which product is used to calculate velocity? In both of these cases, I am assuming that HPLC was used to analyze the reactions, which was not noted in the experimental.

In the intro, it notes that XanO4 is a flavin-dependent monooxygenase from an actinomycetes that catalyzes xanthone formation. Was this work performed *in vitro*? If so, how is FlsO1 different that XanO4. Is the major difference that a Baeyer-Villiger oxygenase reaction was proposed following epoxidation of XanO4, but shown to not be the case for FlsO1? It would be helpful to have a better description of XanO4 in line 193-195 to highlight what advances have been made with FlsO1. This has relevance with respect to the novelty of the presented work.

Grammatical errors include the following:

1. Line 45: 'were limited' should be 'are limited' or 'have been limited'

2. Line 54 and 55: delete 'Recent years' and change the end of the sentence to 'largely expanded in recent years by...'
3. Line 74, suggest deleting 'serendipitous'
4. Line 77, suggest deleting 'unequivocally'
5. Line 79: suggest deleting "In the meanwhile"
6. Line 91: change 'in near' to 'to near'
7. Line 91: suggest change to "were independently incubated with PJM..."
8. Line 119: change "their further" to "any"
9. Line 129: change "prone to spontaneously become 16" to "spontaneously converted to 16"
10. Line 138: delete "In the meanwhile"
11. Line 153 and 154: change "Several rounds of tough efforts afforded" to "Despite exhaustive efforts, only 0.4 mg of 22 was obtained due..."
12. Line 169 and 170: change "lactone ring and a following decarboxylation" to "lactone ring with concomitant decarboxylation"
13. Lines 161-179: the mechanism is described in past tense (was). Perhaps this is better described as in the present, like "we propose Fls01 first catalyzes hydroxylation..." etc.
14. Line 183. Suggest delete "Hence, the"
15. Line 184: sources should be source
16. Line 213: "mediated" incorrectly spelled
17. Line 225: change "for their" to "because of their"
18. Line 234: should L-cystine be L-cysteine?
19. Line 242: methylations is incorrectly spelled
20. Lines 289: what is meant by decreased activities? What product/s was/were detected? Which product was used to determine activity?
21. Line 303: change to "Amino acid sequence"
22. Line 323: change to "functional diversity"
23. Figure 3: xanthone incorrectly spelled in figure part 1

Response to reviewers of NCOMMS-21-49474

Reviewer #1 (Remarks to the Author):

The manuscript of Yang et al. provides new insights in the way a single monooxygenase is able to form the xanthone moiety in bacterial secondary metabolites. Except for thorough chemical analyses, also details on the enzyme structure and mechanism are reported. The study has been carefully performed, and contains many interesting data.

While the results are novel, there is already quite some information on related flavoprotein monooxygenases, particularly those involved in microbial secondary metabolite pathways. Especially, the work performed on the XanO4 contains already leads for how such enzyme can form a xanthone. In that context, this manuscript is somewhat confirmative, building on previous results and hypotheses.

Comment 1.1. *Also, the observed products and intermediates still cannot rule out alternative mechanisms. The proposed scheme in Figure 3a contains several intermediates that have not been experimentally verified.*

Response 1.1. We thank reviewer #1 for the insightful comments. We agree that the proposed scheme in original Fig. 3a contains several intermediates lacking experimental verification, especially for those intermediates involved the proposed Baeyer-Villiger oxidation step. FlsO1 is proposed to catalyze three oxidation steps. The first two steps, namely, the hydroxylation at C-12 and the epoxidation at C-6a/C-12a, have been experimentally confirmed by structurally identifying reaction intermediates/products (such as **19** and **22**) or their derivatives (such as **24** and **25**). However, the involvement of a Baeyer-Villiger oxidation reaction as the third step is still highly hypothetical for lacking experimental verification. Here, the optimization of the FlsO1 reaction conditions (200 μ M **8**, 10 μ M FlsO1, 1.5 mM NADPH in 50 mM phosphate buffer, pH 6, upon incubation at 30 °C for 15-20 min) led to the isolation of another two new structures **26** and **27**, which were identified to be C-ring opened derivatives with a methyl ester unit attached at C-6a and an epoxy group at C-6a/C-12a. The structure characterization of **26** and **27** supports the presence of a lactone ring-containing reaction intermediate **29**, which was probably generated by a Baeyer-Villiger oxidation of **22**.

Comment 1.2. *While the data are convincing for the first two oxygenations, the third seems a bit out of place; it would involve a Baeyer-Villiger oxidation. This is at odds with the first two steps that would rely on a hydroperoxyflavin intermediate, the BV oxidation is supposed to be catalyzed by a peroxyflavin intermediate. No explanation is given how/why the enzyme would be able to form two different enzyme intermediates.*

Response 1.2. We appreciate reviewer #1 for raising concerns on the formation of two

different enzyme intermediates, e.g. a hydroperoxyflavin or a peroxyflavin. Usually, the reduced flavin reacts with oxygen through a radical mechanism, resulting in two reactive intermediates C4a-hydroperoxyflavin (FADOOH) or C4a-peroxyflavin (FADOO⁻). The formation of the two intermediates is dependent on the protonation state of the distal —OOH. C4a-hydroperoxyflavin performs electrophilic attack (for hydroxylation) on the substrate, while C4a-peroxyflavin conducts nucleophilic attack (for epoxidation and Baeyer-Villiger oxidation) on the substrate.

a: the bifunctional FPMO in rifamycin pathway:

b: the bifunctional FPMO in legonmycins pathway:

Supplementary Fig. 78. The proposed involvement of two forms of reactive flavin species hydroperoxyflavin (FADOOH) or a peroxyflavin (FADOO⁻) in a single enzyme-catalyzed reactions in natural product biosynthesis. **(a)** The proposed mechanism of Rif-ORF17 in the biosynthesis of rifamycin; **(b)** the proposed mechanism of LgnC in the biosynthesis of legonmycin.

The explanation for how/why the enzyme would be able to form two different enzyme intermediates is tough. Nonetheless, two reactive flavin intermediates involved in the catalysis of a single FPMO enzyme have been reported in the biosynthesis of different natural product pathways (**revised Supplementary Fig. 78**).

For example, the enzyme Rif-Orf17 performs a phenolic hydroxylation (FADOOH) on C-2 to linearize rifamycin SV, and also conducts a Baeyer-Villiger oxidation (FADOO⁻) on the 1-carbonyl of rifamycin S in rifamycin biosynthetic pathway (**revised ref. 45**; 10.1021/acs.orglett.1c00485). The multifunctional FPMO LgnC has also been reported to mediate an unusual ring expansion/contraction mechanism by catalyzing a Baeyer-Villiger (BV) oxidation (FADOO⁻) and a subsequent stereospecific hydroxylation (FADOOH) in legonmycin biosynthetic pathway (**revised ref. 46**; 10.1002/anie.201502902).

In this work, more DHR (**9**) (a spontaneous oxidation/dehydration product from C-12 hydroxylated intermediate **19** by FADOOH) was observed when the reaction was performed in slightly acidic conditions (*e.g.* in PBS buffers of pH 6.0 and pH 6.5, Fig. 2a); in contrast, much fewer DHR (**9**) was observed when performing the reaction in slightly basic conditions (*e.g.* in PBS buffers of pH 7.0 and pH 7.5, Fig. 2a). These observations may reflect that the micro-environmental protonation or deprotonation status may indeed influence the FADOOH: FADOO⁻ ratio. In that way, more FADOO⁻ might be formed in basic conditions to compete the same substrate (**20**, spontaneous oxidized **19** by O²) for the epoxidation and the Baeyer-Villiger (BV) oxidation, which advances the reaction to form the final reaction product **16**, rather than a spontaneous dehydration of **20** to produce **9**.

Comment 1.3. *In fact, one could imagine that also the third oxygen insertion is a hydroxylation, but then ortho, instead of para in the first hydroxylation (installing an OH 'left' from the carbon that will be removed). Such highly hydroxylated compound can also result in lactone formation. In this context, it is a bit surprising that a recent paper by the Teufel group was not included in the discussion (doi: 10.1002/anie.202109384) where sequence-related bacterial monooxygenases have been reported, including a structure with substrate ligands. Also in that study, hydroxylation leads to ring opening, without the need for a B-V reaction.*

Response 1.3. We sincerely appreciate reviewer #1 for reminding us of the recent fantastic work by the Teufel group that brings a new mechanism for FPMO-catalyzed ring opening reaction. The isolation and identification of two new structures **26** and **27** during the manuscript revision undoubtedly confirmed a C-ring opening reaction by a C-C bond cleavage. We proposed that **27** was derived from an unstable intermediate **29** that was resulted from the FlsO1-catalyzed Baeyer-Villiger oxidation to insert an oxygen at the left of the C-7 keto group of the characterized intermediate **22**. Subsequently, a MeOH-mediated lactone ring opening reaction gave rise to **27**, which was spontaneously dehydrated to **26** (Fig. 3a, pathway c in the following scheme, also shown as **Supplementary Fig. 77**). However, by looking carefully into the ring opening mechanism in the biosynthesis of rubromycin by the Teufel group (pathway a in the following scheme, also shown as **Supplementary Fig. 77**), we realized that an *ortho*-hydroxylation was an alternative mechanism to generate **27** from **22** (pathway b in the following scheme; also shown as **Supplementary Fig. 77**). In this way, a C7, C8- β -dicarbonyl unit is formed by an *ortho*-hydroxylation at C-7a of **22**, and is ready for a

retro-Claisen condensation, by which an attack of H₂O (or MeOH) at the C-7 keto group would trigger the ring opening reaction to yield **30** (or **27**). However, an organic retro-Claisen condensation reaction generally requires base (**revised ref. 43, *Angew. Chem. Int. Ed.*, 47, 7446-7449, 2008**) or Lewis acid catalysts (**revised ref. 44, *Angew. Chem. Int. Ed.*, 46, 7793-7795, 2007**), and would unlikely happen spontaneously. In contrast, the lactone ring in the proposed intermediate **29** is easier to encounter spontaneous hydrolysis or alcoholysis. In this work, the shunt product **27** was simply captured by adding MeOH to terminate the FlsO1 reaction, indicating that the C-ring opening reaction could happen in a mild condition. Taken together, we prefer a Baeyer-Villiger oxidation mechanism for the C–C bond cleavage in the FlsO1 reaction, and the GrhO5-like *ortho*-hydroxylation mechanism is unfavorable.

Supplementary Fig. 77. The comparison of mechanisms for FPMOs-catalyzed ring opening reaction. (a) The proposed ring opening mechanism of GrhO5-catalyzed *ortho*-hydroxylation by the Teufel group. (b) The proposed ring opening mechanism of hydroperoxyflavin mediated hydroxylation followed with retro-Claisen condensation. (c) The proposed ring opening mechanism of peroxyflavin mediated Baeyer-Villiger oxidation.

Comment 1.4. Some text is a bit difficult to follow. For example (in the abstract), in the abstract the terms 'xanthone framework', 'the xanthone formation' and 'confirmative

evidence' should be better introduced/explained.

Response 1.4. Thanks for the comments. We have revised the abstract by changing the words “xanthone framework” to “xanthone-containing”, “of the xanthone formation” to “of the formation of xanthone”, and “confirmative evidence” to “biochemical evidence”.

Comment 1.5. *Also, FMO is not the proper term to use as it is commonly used for a subclass of flavin-containing monooxygenases that are not related to the reported enzymes. Please use FPMOs (flavoprotein monooxygenases) to avoid confusion (original ref for classification: doi: 10.1016/j.jbiotec.2006.03.044).*

Response 1.5. We appreciate the great comments on using the proper term to describe the subclass of enzymes studied in this work. The term “FMO” has been revised to “FPMO” and the original reference was cited as **revised ref. 18**.

Comment 1.6. *Be careful with providing original/proper references: when discussing the PHBH family and in/out position of the FAD cofactor, refs 48&49 are not the most logical, consider doi: 10.1096/fasebj.9.7.7737455.*

Response 1.6. Thanks for raising concerns on citing the original/proper references. The previously-cited references 48 & 49 have been replaced by the new one “doi: 10.1096/fasebj.9.7.7737455 (**revised ref. 56**)” to discuss the PHBH family enzymes and in/out position of the FAD cofactor.

Comment 1.7. *The remark that the group of FPMOs studies in this manuscript is in some way sequence-related to two-component BVMOs and fungal cofactor-independent dioxygenases is totally wrong. The respective Fig S68 should be removed. It would be better to have a complete as possible tree of sequences closely related to FlsO1.*

Response 1.7. In the original submission, we performed bioinformatics analysis of FlsO1-related enzymes in Fig. S68 (original) by similar functions (not merely on amino acid sequence analysis): (1) comparison of FlsO1 with two-component BVMOs for their catalyzing Baeyer-Villiger reactions; (2) comparison of FlsO1 with fungal cofactor-independent dioxygenases (originally annotated also as Baeyer-Villiger enzymes) because these enzymes were involved in mediating the formation of xanthenes with different mechanisms. We realize that such kind of analysis may bring confusions to readers, as being pointed out by reviewer #1. Therefore, we made a sequence-based analysis of FlsO1-related enzymes and constructed a new phylogenetic tree according to the work by the Teufel group for classifying group A FPMOs by forming into two separate clades (**revised Supplementary Fig. 90**). Clearly, FlsO1 is found to belong to type II, in contrast to GrhO5-type I enzymes. Accordingly, the original description in the main text “Phylogenetic analysis reveals that these FMOs are well clustered with

type O Baeyer-Villiger monooxygenases (BVMOs), and was distantly related to bacterial two-component BVMOs and fungal dioxygenases” has been revised to read as,

“Phylogenetic analysis reveals that FlsO1 is well clustered with type II group A FPMOs and thus is separated from the GrhO5-type I enzymes⁴² (Supplementary Fig. 90).”

Reviewer #2 (Remarks to the Author):

The authors presented an interesting study about the versatile flavin-dependent monooxygenase FlsO1. Several different oxidation reactions can be catalyzed by FlsO1, including hydroxylation, epoxidation and Baeyer-Villiger oxidation. Substrate mimic, isotope labeling, crystal structure of FlsO1 and MD simulations were performed well to confirm the biocatalytic mechanisms. I recommend this paper to be published in Nature Commun. after major revisions. Several comments are listed to improve the quality of the paper for the possible publication in Nature Commun.

Major points:

Comment 2.1. *In Fig. 5 def, the orientation of intermediate 22 in the active pocket was different from 8 and 20. FadOOH was closed to C12 in the hydroxylation of compound 8, but to C7 in the Baeyer-Villiger oxidation of 22. According to the structure of 22, two carbonyls (C12 and C7) could be the reactive sites, so the distance of C12 and fadOOH should be given in Fig. 5f. The two reactive sites could lead to the different xanthenes in structure.*

Response 2.1. We appreciate Reviewer #2 for the insightful comments. As shown in the following Figs a and b, we aligned the docked models of **20** and **22**, as well as that of **8**, which revealed that **the orientation of the intermediate 22 in the active pocket was similar to 8 and 20, despite a horizontal shift of 22**. To avoid confusion, we added this information to **Supplementary Fig. 88**, as well as a sentence to the legend of Fig. 5 in the revised manuscript to read as,

“Notably, the intermediate **22** adopts similar orientation as **8** and **20**, despite a horizontal shift”.

Supplementary Fig. 88. Alignments of the docked models of 8, 20 and 22. (a) **20** was aligned with **22**. Note that C-12a atom of **20** overlays with C-7 atom of **22**, consistent with that the two carbons are proposed to be attacked by the peroxyflavin FAD-O-O[•]. (b) The alignments of all three docked models of **8**, **20** and **22** reveal similar orientations in the active pocket, suggesting that the unusual stepwise oxidations on **8**, **20** and **22** only require slight swings of the intermediates in the active site of FlsO1.

The docked model of **22**, as shown in the following figure, reveals that the C-12··O-O-FAD distance is shorter than the C-7··O-O-FAD distance, as pointed out by Reviewer 2. However, an obtuse attacking angle is required for a S_N2 type nucleophilic attack at the carbonyl group, which is known as the “Burgi-Dunitz angle” (*Acc. Chem. Res.*, 16, 153-161, 1983). We believe that a selective attack at C-7, **which is confirmed by the isolation of 26 and 27**, may result from combined effects of distances and angles.

Comment 2.2. Indeed, it is difficult to distinguish the position of oxygen atom in xanthone skeleton. I suggest a crystal structure of 16, 17 or 18 should be added.

Response 2.2. Despite many efforts, we failed to get the crystals of compounds **16**, **17** and **18**. More **16** was thus prepared and purified to get cleaner NMR data. In this way, the newly-provided NMR data of **16** can unequivocally distinguish the position of oxygen atom in the xanthone skeleton. In addition, the structure determination of **26/27** can provide additional information to support the position of oxygen atom in the xanthone skeleton of **16**.

Minor points:

Comment 2.3. *In line 157 and Supplementary Fig. 55, the authors claimed a 1,3-migration in the reaction of 22 to 24. I noticed this process was hypothesized to be catalyzed by H⁺, but performed under neutral condition indeed (Fig. 2c). Ref. 39 reported a rearrangement of N-Acyl Isothioureas. Considering the difference between imine and hydroxyl, I suggest more experiments or references to disclose this process.*

Response 2.3. We appreciate reviewer #2 for raising concerns on the mechanism of DTC-mediated conversion of **22** to **24**. This transformation was hypothesized to involve a 1,3-migration deducing from the structure of **24**. Our original aim of adding DTC to the reaction of FlsO1 is to trap the epoxy intermediate **22** and to confirm the second step of oxidation. Thus, the preparative reaction of FlsO1 with **8** was performed under neutral conditions as uniformly set for FlsO1 reaction, because we used **8** (not using **22** directly) as the substrate for getting **24**. However, after determining the structures of **24**, we realize the occurrence of a 1,3-migration. As suggested by reviewer #2, the process of 1,3-migration was hypothesized to be catalyzed by H⁺. Therefore, we compared the efficiency of FlsO1-catalyzed conversion of **8** to **24** under different conditions with pH values ranging from 6–8. It was observed that the lower of the buffer pH, the more **24** was produced. Surprisingly, only very minor amount of **24** was observed in buffer with pH 8. These data indicated that an acid condition was critical for the production of **24** from **8** and provided indirect support for the involvement of 1,3-migration in the reaction of **22** to **24**. To our delight, the unequivocal presence of an epoxide moiety in the recently identified compounds **26** and **27** (in revised Fig. 3a, Supplementary Figs. 58–71 and Supplementary Table 5) provides strong support for FlsO1-catalyzed epoxidation reaction. The original ref. 39 was deleted since it was not relevant to this study. The proposed mechanism of 1,3-migration of hydroxyl was revised in Supplementary Fig. 56.

Supplementary Fig. 56. The proposed mechanism for the formation of 24. (a) HPLC analysis of FlsO1 enzyme assays. The assays were performed by incubation of 200 μ M **8** in the presence of 2 mM NADPH and 5 mM DTC (**23**): (i) control (no enzyme); (ii) pH 6.0; (iii) pH 7.0; (iv) pH 8.0, (ii–iv) assays with 5 μ M FlsO1. (v) pH 6.0; (vi) pH 7.0; (vii) pH 8.0, (v–vii) assays with 10 μ M FlsO1. The reactions were performed in PBS buffers (50 mM) at 30 $^{\circ}$ C (iii–ix) for 6 min. (b) The proposed mechanism for the formation of **24** from DTC-trapping reaction of **22**, the second step of oxygenation product of **8**.

Comment 2.4. The compound **16** should be further purified, according to its 1H NMR and ^{13}C NMR (Fig. 5, 6 in SI).

Response 2.4. According to the suggestion of reviewer #2, we further purified compound **16** to get cleaner new NMR spectra that were used to replace the old ones (see revised Supplementary Figs. 5–10).

Comment 2.5. I notice that the isotope incorporation in $^{18}O_2$ labeling experiments is significantly lower than 97% (the gas be used), I am curious why it decreased.

Response 2.5. We thank reviewer #2 to raise this question on the $^{18}O_2$ labelling experiments. In fact, the $^{18}O_2$ labelling experiments were not performed strictly under pure $^{18}O_2$ (97%) since the lack of anaerobic equipment. To perform the $^{18}O_2$ labeling experiments, a stream of high-purity nitrogen was first bubbled into the FlsO1 reaction system in an Eppendorf microtube to thoroughly remove the atmospheric O_2 , which

would not be 100% removed. Subsequently, the 97% $^{18}\text{O}_2$ gas was bubbled into the reaction system to initiate the labelling reaction. Because the atmospheric O_2 could not be completely eliminated from the reaction system, the observed amount of labelled $^{18}\text{O}_2$ is less than 97%. Nonetheless, although the $^{18}\text{O}_2$ labeling experiments were not perfectly performed, it could provide us enough qualitative information to estimate the number of ^{18}O -labelled atoms in each product to help understanding the complicated oxidation steps.

Comment 2.6. *The format of references should be unified.*

Response 2.6. According to the suggestion, the format of references have been unified, revised as follows,

In ref. 29, "*Mar. Drugs* **17**, (2019)" is changed to "*Mar. Drugs* **17**, 150 (2019)".

In ref. 30, "gene flsO1" is changed to "gene *flsO1*".

In ref. 31, "Inactivation" is changed to "inactivation" and "gene flsP " is changed to "gene *flsP*"

Reviewer #3 (Remarks to the Author):

The authors report the functional and structural characterization of a flavin-dependent oxygenase FlsO1. FlsO1 is demonstrated in vitro to be able to complement another oxygenase FlsO2 by converting pre-jadamycin B to dehydrorebelomycin b, but also convert pre-jadamycin to two xanthone products. FlsO1's "natural activity" for hydroxylating a benzo[b]-fluorene intermediate was also demonstrated using a surrogate substrate called nenestain C. A substantial amount of data is provided, and structural assignments of enzyme products are assigned by extensive spectroscopic analysis. The manuscript, although dense, is well written and well organized. The results are interesting and will be valuable to the community.

Comment 3.1. *In general the conclusions are supported by the data. The exception is the statement that FlsO1 catalyzes cryptic xanthone formation. In the biosynthetic field, cryptic is often used to describe an unexpected intermediate in a biosynthetic pathway, but the xanthenes produced here are apparently the final product.*

Response 3.1. We appreciate reviewer #3 to explain the meaning of the word "cryptic" in a biosynthetic pathway. According to the suggestion, we have replaced the word "cryptic" in the title to "unexpected". In page 5, the statement "Discovery of FlsO1-mediated **cryptic** xanthone formation" has been revised as "Discovery of FlsO1-mediated unexpected xanthone formation".

Comment 3.2. *More importantly, it's not clear whether the xanthenes have any significance as natural products from this strain. Are xanthenes detected from*

Micromonospora rosaria SCSIO N160 or the *flsO2*-inactivation mutant? It's most likely that xanthone formation is a consequence of *in vitro* conditions wherein the incorrect substrate is provided to the enzyme. This is supported by the kinetic data provided by the authors, which suggests the conversion to xanthenes is not efficient and simply an off-pathway reaction that is a consequence of *in vitro* characterization. It would be useful to note if this mechanism of xanthone formation is actually expected to naturally occurring.

Response 3.2. This work was initially designed to search for the cluster-situated enzyme that could complement the function of FlsO2, which was proven to convert PJM (**8**) to DHR (**9**); however, minor amount of DHR (**9**) was still detected in the $\Delta flsO2$ mutant (**revised ref. 27**, 10.1021/acs.orglett.5b02683). We confirmed in this work that FlsO1 could indeed transform PJM (**8**) to DHR (**9**) *in vitro*. This fact indicated that the trace production of DHR (**9**) was probably accounted for the *in vivo* C-12 hydroxylation of **8** by FlsO1, or in other words, FlsO1 could indeed complement the *in vivo* function of FlsO2. However, we agree with reviewer #2 that it's most likely that xanthone formation is a consequence of *in vitro* conditions wherein the incorrect substrate is provided to the enzyme FlsO1 since the xanthone **16** is only significantly produced when using high concentration of FlsO1 in the assays (**revised Supplementary Fig. 79**). We then carefully analyzed and compared the products of the wild type *Micromonospora rosaria* SCSIO N160 and the $\Delta flsO2$ mutant. To our delight, a very tiny amount of fluoxanthone A (**16**) was also detected in the $\Delta flsO2$ mutant, which was demonstrated by LC-MS analysis and comparison of its UV spectrum and retention time with standard **16** (**revised Supplementary Fig. 80**). This observation suggests that the mechanism proposed in this work for xanthone formation is expected to occur naturally.

Comment 3.3. *Regarding the kinetic data collected with FlsO1 and FlsO2 with compound 8: what is the product that is being detected to calculate the velocity? Based on the HPLC trace in Figure 2a at pH 7, it appears that very little 9 is being formed, while 16 and 18 are relatively abundant. Is the calculated velocity through detecting 9? How does the kinetic analysis with respect to 16 and 18 compare? Similarly, the HPLC analysis of the FlsO2 shows that 10 is formed in greater amounts than 9. Which product is used to calculate velocity? In both of these cases, I am assuming that HPLC was used to analyze the reactions, which was not noted in the experimental.*

Response 3.3. We appreciate reviewer #3 for insightful comments on determining the kinetic parameters of FlsO1 and FlsO2 with compound **8**. In the original submission, the velocity was calculated upon all products that were detected from the HPLC analysis. The difference was that the enzyme concentrations used for the reactions (5 μ M for FlsO1 vs 0.25 μ M for FlsO2). Actually, under the assay conditions, **9** is the sole product for FlsO2. In **Fig. 2a (trace ii)**, we deliberately carried out the FlsO2 reaction in 16° C (please see **the legend for Fig. 1**) to clearly show the presence of the

intermediate **10**. FlsO2 is a bifunctional enzyme that can also catalyze the dehydration reaction after C-12 hydroxylation, leading to **10**. In this regard, we intend to highlight that the mechanisms of FlsO1- and FlsO2-catalyzed conversion of **8** to **9** are different in terms of the accumulated intermediates (**10** for FlsO2, while **19/20** for FlsO1). To avoid the confusion, we added an additional HPLC trace to show the almost complete conversion of **8** to **9** by FlsO2.

Upon concerns of reviewer #3, we realized that the high concentration of FlsO1 used for kinetic analysis may bring “false” outcomes. In other words, the different UV spectra of **9** and **16/18** would also cause discrepancy in obtaining the “real” results. Therefore, to get more reliable kinetic results, we tested the FlsO1 reaction with **8** under lower concentrations of FlsO1 in a short time. To our delight, **9** was detected as the only product in assays with 0.1-0.5 μM FlsO1 in 0-12 min at 30° C. Since very few products could be detected with higher concentration of the substrate **8** when using FlsO1 lower than 0.5 μM , we chose to measure the FlsO1 kinetic parameters under the following conditions: PJM (**8**) was set at the concentrations of 15, 25, 50, 75, 100, 150, 200, 250, 300, 400, 1000, and 1500 μM ; enzyme assays were performed in triplicates in 50 mM phosphate buffer (pH 7.0) with **0.5 μM** FlsO1 and 2 mM NADPH, by incubation at 30 °C for 4 min (**revised Supplementary Fig. 79**).

Comment 3.4. *In the intro, it notes that XanO4 is a flavin-dependent monooxygenase from an actinomyces that catalyzes xanthone formation. Was this work performed in vitro? If so, how is FlsO1 different that XanO4. Is the major difference that a Baeyer-Villiger oxygenase reaction was proposed following epoxidation of XanO4, but shown to not be the case for FlsO1? It would be helpful to have a better description of XanO4 in line 193-195 to highlight what advances have been made with FlsO1. This has relevance with respect to the novelty of the presented work.*

Response 3.4. The function of the flavoprotein monooxygenase XanO4 was previously biochemically characterized in vitro (reference 13, *Cell Chem. Biol.* **23**, 508-516, 2016). The main conclusion is that XanO4 catalyzes multistep reactions including BV oxidation, decarboxylation, and oxidative demethoxylation, to convert an anthranquinone to a xanthone (**revised Supplementary Fig. 76**); however, the reaction mechanism was highly hypothetical and was proposed based only on the $^{18}\text{O}_2$ isotopic labeling results of the end product. The detailed process for how XanO4 transforms an anthraquinone to a xanthone and which kind of intermediates are formed are still a black box.

In this work, the flavoprotein monooxygenase FlsO1 was found to convert an angucyclinone precursor PJM (**8**) to a xanthone product (**16**) via three steps of oxidation, including a C-12 hydroxylation, a C-6a/C-12a epoxidation and a Baeyer-Villiger oxidation. The first step of C-12 hydroxylation was absent in the proposed XanO4-catalyzed reaction. We provided detailed biochemical evidence for each step of FlsO1-catalyzed oxidations by identifying reaction intermediates or intermediate-derived products. What's more, how the reaction intermediate **18** was spontaneously transformed to the end xanthone product **16** was clearly elucidated. In addition, a

second way of xanthone formation was proposed by an intermolecular attack of a hydroxyl group to an epoxide (**route 2 in Fig. 3a**).

Although both FlsO1 and XanO4 can catalyze the formation of xanthone products, FlsO1 only shows 23.7% identity in the amino acid sequence to that of XanO4.

Comment 3.5. *Grammatical errors include the following:*

1. *Line 45: 'were limited' should be 'are limited' or 'have been limited'*

Response: In line 45, “were limited” has been replaced by “have been limited”.

2. *Line 54 and 55: delete 'Recent years' and change the end of the sentence to 'largely expanded in recent years by...'*

Response: The original sentence “Recent years, the structure diversity of FST-related angucyclines and angucyclinones was largely expanded by...” is changed to “The structure diversity of FST-related angucyclines and angucyclinones was largely expanded in recent years by...”

3. *Line 74, suggest deleting 'serendipitous'*

Response: The word “serendipitous” has been deleted.

4. *Line 77, suggest deleting 'unequivocally'*

Response: The word “unequivocally” has been deleted.

5. *Line 79: suggest deleting 'In the meanwhile'*

Response: The phrase “In the meanwhile” has been deleted.

6. *Line 91: change 'in near' to 'to near'*

Response: The expression “in near” is changed to “to near”.

7. *Line 91: suggest change to "were independently incubated with PJM..."*

Response: The sentence “Subsequently, FlsO1, FlsO3, FlsO4 and FlsO5 were incubated respectively with PJM (8)...” has been modified to “Subsequently, FlsO1, FlsO3, FlsO4 and FlsO5 were independently incubated with PJM(8)...”

8. *Line 119: change "their further" to "any"*

Response: The expression “their further” has been changed to “any”.

9. Line 129: change “prone to spontaneously become 16” to “spontaneously converted to 16”

Response: The expression “prone to spontaneously become 16” is changed to “spontaneously converted to 16”.

10. Line 138: delete “In the meanwhile”

Response: The phrase “In the meanwhile” has been deleted.

11. Line 153 and 154: change “Several rounds of tough efforts afforded” to “Despite exhaustive efforts, only 0.4 mg of 22 was obtained due...”

Response: The sentence “Several rounds of tough efforts afforded only 0.4 mg of 22 due to...” has been changed to “Despite exhaustive efforts, only 0.4 mg of 22 was obtained due to...”.

12. Line 169 and 170: change “lactone ring and a following decarboxylation” to “lactone ring with concomitant decarboxylation”

Response: Since the ring-opened product (30 in Fig. 3) is numbered in the revised manuscript, the sentence is now read as,

“In the FlsO1 reaction cascade, a hydrolytic opening of the lactone ring in 29 affords 30 and a subsequent decarboxylation yields the putative intermediate 31.”

13. Lines 161-179: the mechanism is described in past tense (was). Perhaps this is better described as in the present, like “we propose FlsO1 first catalyzes hydroxylation...” etc.

Response: According to the suggestion, the mechanism is described in present tense.

14. Line 183. Suggest delete “Hence, the”

Response: The expression “Hence, the” has been deleted.

15. Line 184: sources should be source

Response: The word “sources” is changed to “source”.

16. Line 213: “mediated” incorrectly spelled

Response: The incorrectly spelled word “meidated” has been changed to “mediated”.

17. Line 225: change “for their” to “because of their”

Response: The expression “for their” is changed to “because of their”.

18. Line 234: should L-cystine be L-cysteine?

Response: The incorrectly spelled “L-cystine” has been replaced by “L-cysteine”.

19. Line 242: methylations is incorrectly spelled

Response: The incorrectly spelled word “methlations” has been revised to “methylations”.

20. Lines 289: what is meant by decreased activities? What product/s was/were detected? Which product was used to determine activity?

Response: The expression “decreased activities” is intended to mean that the mutants H76F, R95A, V215G, F258A and M284A display less consumption of the substrate **8** than the wild type FlsO1, indicating declined catalytic properties. To clearly express the idea, we change the sentence “The mutants H76F, R95A, V215G, F258A and M284A displayed decreased activities, which may result from the weakened substrate binding.....” to “The mutants H76F, R95A, V215G, F258A and M284A displayed decreased activities, judging from their ability by consuming less **8** than the wild type FlsO1, which may result from their weakened substrate binding ability.”

21. Line 303: change to “Amino acid sequence”

Response: The expression “Amino sequence” has been revised to “Amino acid sequence”.

22. Line 323: change to “functional diversity”

Response: The expression “function diversity” is changed to “functional diversity”.

23. Figure 3: xhanthone incorrectly spelled in figure part 1

Response: In Figure 3, the incorrectly spelled word “xhanthone” in figure part 1 has been changed to “xanthone”.

REVIEWER COMMENTS

Reviewer #1 (Remarks to the Author):

The manuscript has matured by, among others: some rewriting, providing alternative explanations for the C-C bond cleavage and a better description of the studied enzyme, FlsO1. The claims are now better explained. and are supported by the presented data. The data and insights are novel and of interest for the field.

Minor comment: the phylogenetic analysis / sequence alignment shown in Fig S90 should include the prototype p-hydroxybenzoate hydroxylase. While it is even mentioned in the main text, it is missing in the phylogenetic tree.

Reviewer #2 (Remarks to the Author):

The authors have adequately addressed my comments through their revisions to the text, and have made a number of other improvements based on the advice of the other reviewers. These results are likely to be of great interest to biosynthesis and biochemists. I therefore recommend that this manuscript be published without further revisions.

Reviewer #3 (Remarks to the Author):

The revised version of the manuscript is greatly improved and all prior concerns were satisfactorily addressed.

Line 274: should be L-cysteine instead of L-cystine?

Response to Reviewers' Comments (NCOMMS-21-49474A)

Reviewer #1 (Remarks to the Author):

The manuscript has matured by, among others: some rewriting, providing alternative explanations for the C-C bond cleavage and a better description of the studied enzyme, FlsO1. The claims are now better explained. and are supported by the presented data. The data and insights are novel and of interest for the field.

Minor comment: *the phylogenetic analysis / sequence alignment shown in Fig S90 should include the prototype *p*-hydroxybenzoate hydroxylase. While it is even mentioned in the main text, it is missing in the phylogenetic tree.*

Response: We appreciate the positive comments from reviewer #1 and the further suggestions on revising our phylogenetic analysis / sequence alignment of FlsO1. As suggested, the prototype *p*-hydroxybenzoate hydroxylases (PHBHs) have been added to the phylogenetic tree in the revised Supplementary Fig 90.

Reviewer #2 (Remarks to the Author):

The authors have adequately addressed my comments through their revisions to the text, and have made a number of other improvements based on the advice of the other reviewers. These results are likely to be of great interest to biosynthesis and biochemists. I therefore recommend that this manuscript be published without further revisions.

Response: We appreciate the encouraging comments from reviewer #2. No further revisions are requested by Reviewer #2.

Reviewer #3 (Remarks to the Author):

The revised version of the manuscript is greatly improved and all prior concerns were satisfactorily addressed.

Minor comment: *Line 274: should be L-cysteine instead of L-cystine?*

Response: We appreciate reviewer #3 for the careful reading of our manuscript. In line 274, the “L-cysteine” was mistyped as “L-cystine”. We have corrected this in the revised text.

REVIEWERS' COMMENTS

Reviewer #1 (Remarks to the Author):

The authors have adequately addressed my comment by preparing a new figure. I recommend publication without further revisions.

Response to Reviewer Comments (NCOMMS-21-49474B)

Reviewer #1 (Remarks to the Author):

Comment:

The authors have adequately addressed my comment by preparing a new figure. I recommend publication without further revisions.

Response: We are glad that no further revisions are requested by Reviewer #1.